# Transverse tubule remodeling enhances Orai1-dependent Ca$^{2+}$ entry in skeletal muscle

Antonio Michelucci[1,2], Simona Boncompagni[2,3], Laura Pietrangelo[2,3], Maricela García-Castañeda[1], Takahiro Takano[1], Sundeep Malik[1], Robert T Dirksen[1†*], Feliciano Protasi[2,4†]

[1]Department of Pharmacology and Physiology, University of Rochester School of Medicine and Dentistry, Rochester, United States; [2]Center for Research on Ageing and Translational Medicine (CeSI-MeT), University Gabriele d'Annunzio, Chieti, Italy; [3]Department of Neuroscience, Imaging and Clinical Sciences (DNICS), University Gabriele d'Annunzio, Chieti, Italy; [4]Department of Medicine and Ageing Sciences (DMSI), University Gabriele d'Annunzio, Chieti, Italy

**Abstract** Exercise promotes the formation of intracellular junctions in skeletal muscle between stacks of sarcoplasmic reticulum (SR) cisternae and extensions of transverse-tubules (TT) that increase co-localization of proteins required for store-operated Ca$^{2+}$ entry (SOCE). Here, we report that SOCE, peak Ca$^{2+}$ transient amplitude and muscle force production during repetitive stimulation are increased after exercise in parallel with the time course of TT association with SR-stacks. Unexpectedly, exercise also activated constitutive Ca$^{2+}$ entry coincident with a modest decrease in total releasable Ca$^{2+}$ store content. Importantly, this decrease in releasable Ca$^{2+}$ store content observed after exercise was reversed by repetitive high-frequency stimulation, consistent with enhanced SOCE. The functional benefits of exercise on SOCE, constitutive Ca$^{2+}$ entry and muscle force production were lost in mice with muscle-specific loss of Orai1 function. These results indicate that TT association with SR-stacks enhances Orai1-dependent SOCE to optimize Ca$^{2+}$ dynamics and muscle contractile function during acute exercise.

DOI: https://doi.org/10.7554/eLife.47576.001

*For correspondence:
robert_dirksen@urmc.rochester.edu

†These authors contributed equally to this work

**Competing interests:** The authors declare that no competing interests exist.

## Introduction

Store-operated Ca$^{2+}$ entry (SOCE), a Ca$^{2+}$ entry mechanism activated by depletion of intracellular stores, is mediated by the coupling between stromal interacting molecule-1 (STIM1), a luminal Ca$^{2+}$ sensor protein in the endoplasmic reticulum (ER) membrane (*Roos et al., 2005*), and Ca$^{2+}$-permeable Orai1 channels in the plasma membrane (*Feske et al., 2006*). In skeletal muscle, SOCE is similarly coordinated by a functional interaction between STIM1 in the sarcoplasmic reticulum (SR) and Orai1 in the transverse tubule (TT) (*Kurebayashi and Ogawa, 2001*; *Lyfenko and Dirksen, 2008*; *Stiber et al., 2008*), specialized invaginations of the surface membrane that propagate action potentials to trigger SR Ca$^{2+}$ release during excitation-contraction (EC) coupling. While the precise physiological role(s) of SOCE in muscle remains incompletely defined, SOCE has been proposed to promote myoblast fusion/differentiation (*Darbellay et al., 2009*) and muscle development (*Kurebayashi and Ogawa, 2001*; *Wei-Lapierre et al., 2013*; *Carrell et al., 2016*), reduce fatigue during periods of prolonged stimulation (*Wei-Lapierre et al., 2013*; *Carrell et al., 2016*; *Pan et al., 2002*; *Boncompagni et al., 2017*), as well as serve as a counter-flux to Ca$^{2+}$ loss across the TT system during EC coupling (*Koenig et al., 2018*). Importantly, mutations in the genes that encode STIM1 and Orai1 underlie an array of disorders with clinical myopathy as central defining

component. SOCE dysfunction also contributes to the pathogenesis of several other muscle disorders (*Michelucci et al., 2018*). Thus, SOCE plays an important role in both normal muscle development/function and muscle disease.

In resting mammalian skeletal muscle, STIM1 proteins are found throughout the I band of the sarcomere (*Stiber et al., 2008*; *Wei-Lapierre et al., 2013*; *Boncompagni et al., 2017*). On the other hand, Orai1 channels at rest are found almost exclusively at the A-I band junction, in accordance with their localization within the TT membrane of the triad (*Wei-Lapierre et al., 2013*; *Boncompagni et al., 2017*). A modest degree of STIM1-Orai1 co-localization at the A-I band junction at rest has been suggested to explain in part the uniquely rapid activation of SOCE (<1 s) observed in skeletal muscle (*Koenig et al., 2018*; *Launikonis and Ríos, 2007*; *Edwards et al., 2011*). However, STIM1 and Orai1 localization in muscle is not static, as acute treadmill exercise drives a significant reorganization of the sarcotubular membranes in the I band of *extensor digitorum longus* (EDL) fibers of wild type (WT) mice that involves: i) remodeling of the SR into stacks of flat cisternae; ii) elongation of the TT into the I band; and iii) increased co-localization of Orai1 with STIM1 in the I band (*Boncompagni et al., 2017*). This exercise-induced remodeling leads to the formation of new SR-TT contacts that are structurally distinct from triad junctions in that they are oriented longitudinally at the I band (while triads are transversely oriented at the A-I band junction) and display a junctional gap of only 7–8 nm (while triads exhibit a 12 nm junctional gap) (*Boncompagni et al., 2017*). Importantly, the formation of these junctions following acute exercise correlates with an increased resistance to EDL muscle contractile decline during repetitive stimulation in presence of extracellular $Ca^{2+}$, but not under conditions that reduce SOCE (0 $Ca^2$, BTP-2, 2-APB) (*Boncompagni et al., 2017*). As the formation of the SR-TT junctions within the I band following acute exercise are coincident with enhanced STIM1-Orai1 co-localization and force production during sustained activity, these structural elements were suggested to function as *Calcium Entry Units* (CEUs).

In spite of these findings, the stability of CEUs formed following exercise, process for disassembly of the components, and impact on Orai1-dependent SOCE, $Ca^{2+}$ dynamics, and muscle force production remain unknown. In fact, the precise degree to which CEUs formed after exercise promote SOCE, optimize $Ca^{2+}$ dynamics and contribute to the adaptation of muscle function in response to acute exercise is unknown. Here, we addressed these issues using a combination of structural (electron microscopy, EM) and functional ($Mn^{2+}$ quench of fura-2 fluorescence, intracellular $Ca^{2+}$ measurements, and muscle contractility) approaches in EDL muscle and *flexor digitorum brevis* (FDB) fibers from WT and Orai1-deficient mice in the absence of exercise (control) or <1, 6, and 24 hr after acute treadmill exercise. Together, the results support the idea that TT association with SR stacks in the I band promotes Orai1-dependent SOCE needed to replenish releasable $Ca^{2+}$ stores, maintain SR $Ca^{2+}$ release, and maximize contractile force during repetitive stimulation of fast twitch skeletal muscle.

## Results

### SR-stacks increase for up to 6 hr after exercise

We previously reported that EDL muscle fibers from mice subjected to a single bout of incremental treadmill running exhibit a remodeling of SR membranes into flat, parallel stacks of flat cisternae (*SR-stacks*) (*Boncompagni et al., 2017*). To determine their stability, we quantified SR-stacks in EM cross sections of EDL muscles fixed at different time points (<1, 6 and 24 hr) following acute treadmill exercise (*Figure 1A–D*). Shortly after exercise (<1 hr), the percentage of muscle fibers with SR-stacks in the I band (*Figure 1E*), the number of stacks/100 $\mu m^2$ (*Figure 1F*), and SR-stack length (*Figure 1G*) were all significantly increased compared to that of non-exercised control mice. The percentage of fibers with SR-stacks at the I band and the number of SR stacks/100 $\mu m^2$ further increased 6 hr after exercise. All three parameters decreased to values closer to that of fibers from non-exercised control mice 24 hr after exercise. A similar time course was also observed in FDB muscle fibers (*Figure 1—figure supplement 1*).

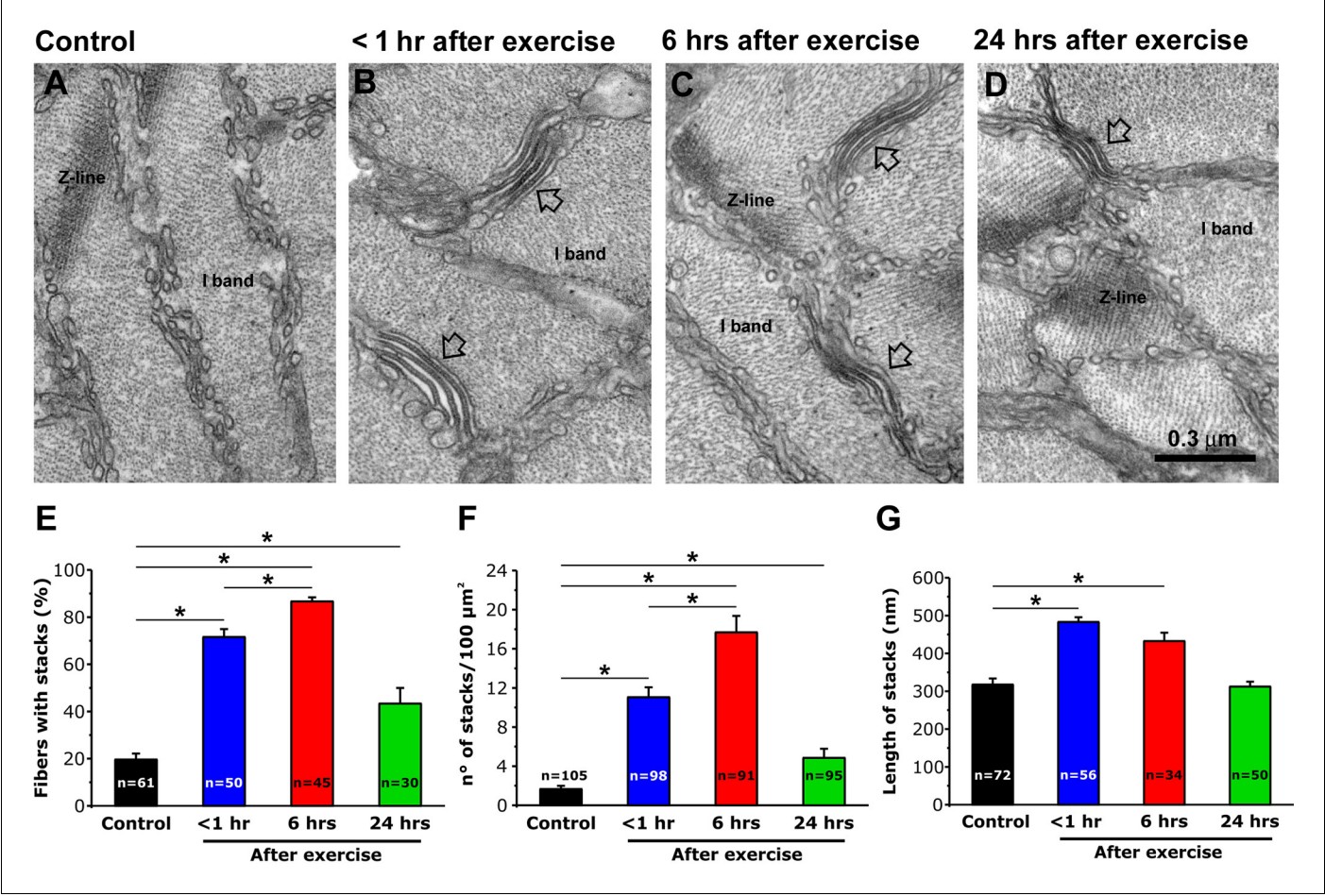

**Figure 1.** SR-stack incidence, number/area, and length after exercise. (A–D) Representative EM images of cross sections (in proximity of I band) of EDL muscle fibers from control mice (A) and from mice < 1 hr (B), 6 hr (C), and 24 hr (D) after acute treadmill exercise (empty arrows point to stacks of SR membranes). (E) Percentage of muscle fibers exhibiting SR-stacks. (F) Number of SR stacks per 100 $\mu m^2$ of cross sectional area. (G) SR-stack length. Numbers in bars (n) indicate the number of fibers analyzed. Number of mice used: Control, n = 4; <1 hr after exercise, n = 4; 6 hr after exercise, n = 4; 24 hr after exercise, n = 3; *p<0.05. Data are shown as mean ± SEM.
DOI: https://doi.org/10.7554/eLife.47576.002

The following figure supplement is available for figure 1:

**Figure supplement 1.** SR-stack incidence, number/area, and T-tubule length after exercise.
DOI: https://doi.org/10.7554/eLife.47576.003

## TT length is increased <1 hr after exercise

Acute exercise also promotes an elongation of the TT from the triad into the I band to form new contacts with stacks of SR cisternae (*Boncompagni et al., 2017*). Thus, we quantified the length of contacts between TT extensions and flat-parallel stacks of SR within the I band at rest, as well as <1, 6, and 24 hr following acute exercise, in samples post-fixed in the presence of ferrocyanide (*Figure 2A–D*). The results indicate that the TT/SR contact length within the I band region of the sarcomere increased 5-fold < 1 hr after exercise and then returned to levels not significantly different from control both 6 and 24 hr after exercise (*Figure 2E*). The increase in TT/SR contact length after exercise is consistent with our prior demonstration (using immunofluorescence and immuno-gold electron microscopy) of increased Orai1-STIM1 co-localization within the I band <1 hr after exercise (*Boncompagni et al., 2017*).

In addition, total TT length/100 $\mu m^2$ was also significantly increased in EDL muscle fibers from mice < 1 hr after exercise (*Figure 2B and F*) compared to that of control samples (*Figure 2A and F*). Importantly, in contrast to that observed for SR-stacks (*Figure 1*), increased extensions of the TT

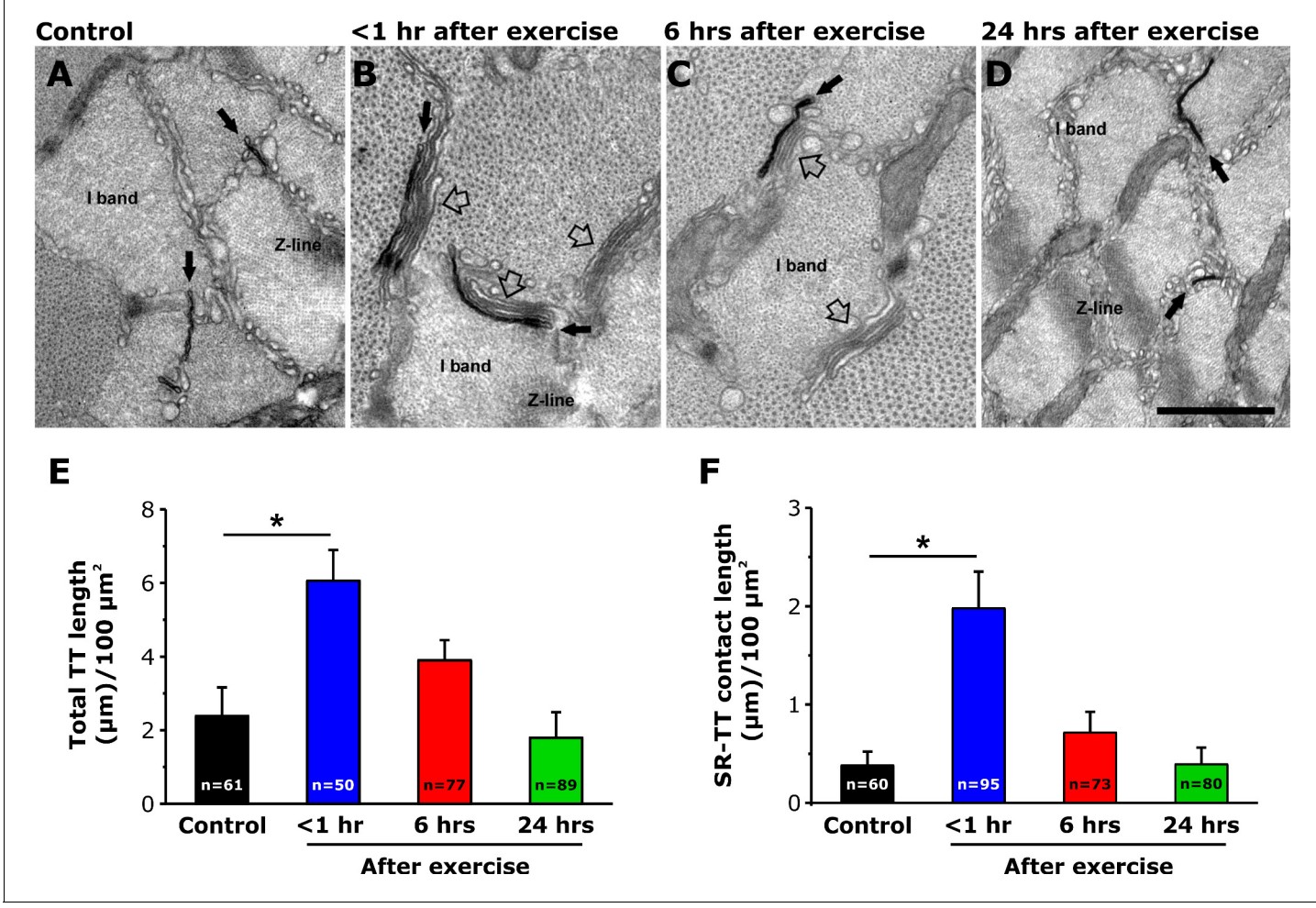

**Figure 2.** TT extension at the I band after exercise. (A–D) Representative EM images of cross sections (in proximity of the I band) of EDL muscle fibers from control mice (A) and from mice < 1 hr (B), 6 hr (C), and 24 hr (D) after acute treadmill exercise (empty arrows point to SR-stacks, while black arrows point to TT extensions which are stained in black with ferrocyanide precipitate). (E and F) Quantitative analysis of TT-SR contact length (E) and total TT length (F) within the I band ($\mu$m/100 $\mu$m² of cross sectional area). Numbers in bars (n) indicate the number of EDL fibers analyzed. Number of mice used: Control, n = 4; <1 hr after exercise, n = 4; 6 hr after exercise, n = 4; 24 hr after exercise, n = 3; *p<0.05. Data are shown as mean ± SEM.

DOI: https://doi.org/10.7554/eLife.47576.004

network (and TT/SR contact length) within the I band, observed <1 hr after exercise, were not different from control 6 and 24 hr after exercise (*Figure 2E and F*). A transient increase in total TT length/100 $\mu$m² and with a more prolonged increase in SR stacks at the I band after exercise was also observed in FDB muscle fibers (*Figure 1—figure supplement 1*).

## Mn²⁺Quench of fura-2 fluorescence is increased <1 hr after exercise

Immunofluorescence and immunogold studies found that exercise-induced formation of CEUs promoted an increase of STIM1 and Orai1 co-localization (*Boncompagni et al., 2017*). To determine the degree to which these structural changes correlate with STIM1-Orai1 function, the maximum rate of Mn²⁺ quench of fura-2 fluorescence was used as an index of Ca²⁺ entry as described previously (*Wei-Lapierre et al., 2013*). Mn²⁺ quench experiments were performed in single FDB fibers both following SR Ca²⁺ store depletion (+depletion; 1 hr incubation with thapsigargin plus cyclopiazonic acid) and in the absence of pharmacological store depletion (-depletion). Although fura-2 emission was unaltered upon application of extracellular Mn²⁺ in non-depleted fibers (*Figure 4—figure supplement 1*), a significant increase in quench was observed in fibers from control mice following store depletion (*Figure 3A*). Importantly, compared to that of control mice, the maximum rate of

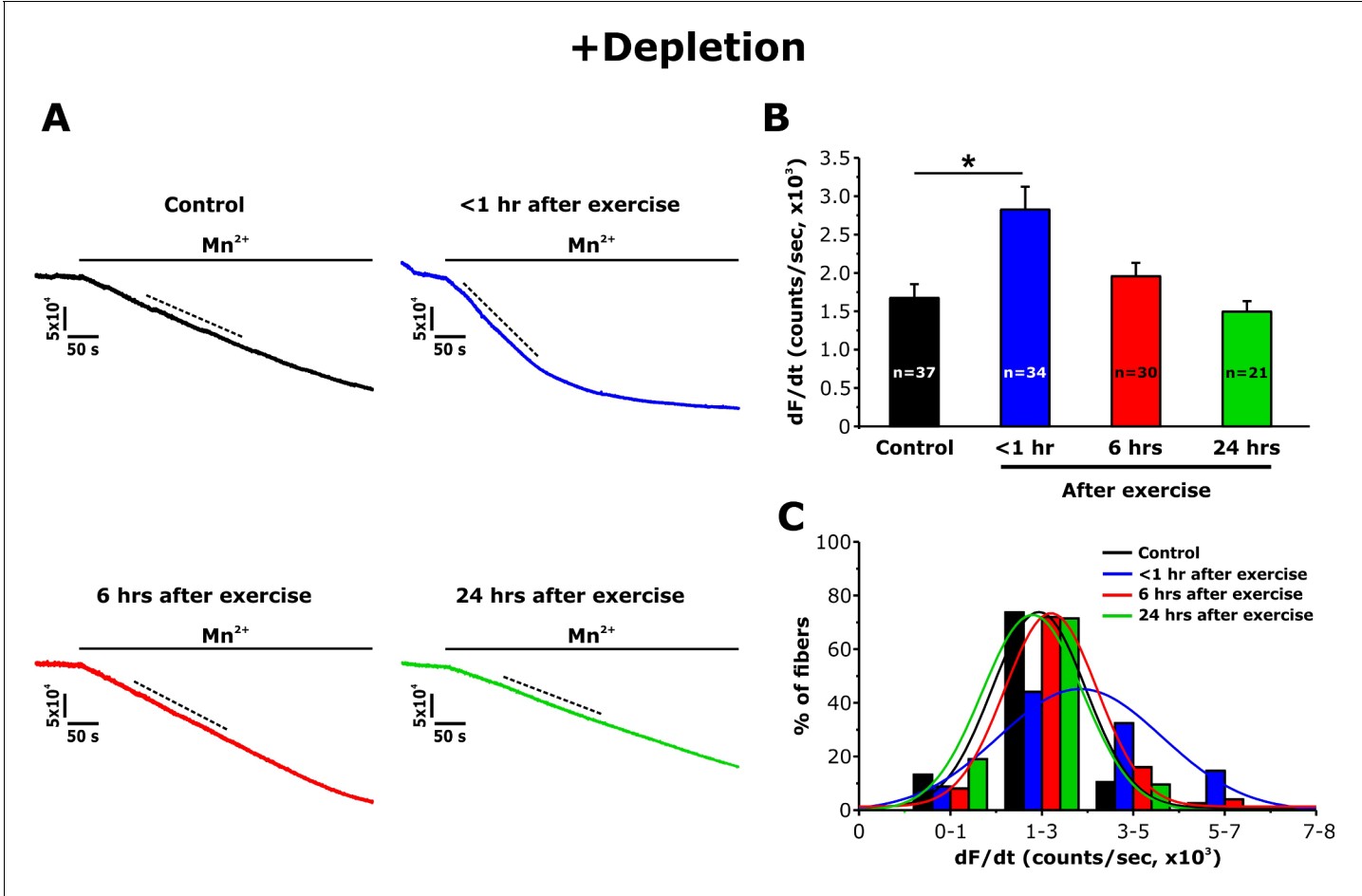

**Figure 3.** Effect of exercise on the maximum rate of Mn²⁺ quench following store depletion. (A) Representative traces of fura-2 fluorescence during application of 0.5 mM Mn²⁺ recorded in FDB fibers following store depletion with 1 µM thapsigargin + 15 µM cyclopiazonic acid (+depletion) isolated from control mice and from mice < 1 hr, 6 hr, and 24 hr after acute treadmill exercise. (B) Quantitative analysis of the maximum rate of Mn²⁺ quench following store depletion. (C) Frequency histogram of percentage of fibers exhibiting different levels of maximal rate of Mn²⁺ quench. Histogram data were fit with a single Gaussian distribution. Numbers in bars (n) reflect the number of FDB fibers analyzed. Number of mice used: Control, n = 5; <1 hr after exercise, n = 5; 6 hr after exercise, n = 3; 24 hr after exercise, n = 3; *$p < 0.05$. Data are shown as mean ± SEM.

DOI: https://doi.org/10.7554/eLife.47576.005

Mn²⁺ quench was significantly increased (~70%) in FDB fibers isolated from mice < 1 hr after treadmill exercise (*Figure 3B*). The maximum rate of Mn²⁺ quench was not different from that of control mice following 6 and 24 hr of recovery. An analysis of the frequency distribution of maximal rate of Mn²⁺ quench in SR depleted fibers showed a shift toward larger rates of quench in fibers from mice < 1 hr after exercise, but not in fibers isolated at 6 and 24 hr of recovery (*Figure 3C*).

Since acute exercise increases STIM1 and Orai1 co-localization within the I band (*Boncompagni et al., 2017*), we also quantified Mn²⁺ quench in FDB fibers in the absence of pharmacological store depletion (-depletion). While fura-2 quench did not occur upon addition of Mn²⁺ to fibers from non-exercised mice, a marked increase in quench rate was observed in fibers from mice < 1 hr after exercise (*Figure 4A*). Consistent with the morphological observations in TT length (*Figure 2*), the maximum rate of Mn²⁺ quench in the absence of pharmacological store depletion was significantly increased at <1 hr after exercise, reduced at 6 hr, but essentially undetectable at 24 hr after exercise (*Figure 4B*). As a result, a shift toward larger Mn²⁺ quench rates was observed in fibers isolated from mice < 1 hr after exercise (*Figure 4C*). Consistent with the Mn²⁺ quench observed in SR Ca²⁺ replete fibers from mice < 1 hr after exercise being mediated by Orai1 channels, Mn²⁺ quench in these fibers was rapidly blocked by an inhibitor of Orai1-dependent SOCE (10 µM BTP-2; *Figure 4—figure supplement 1*).

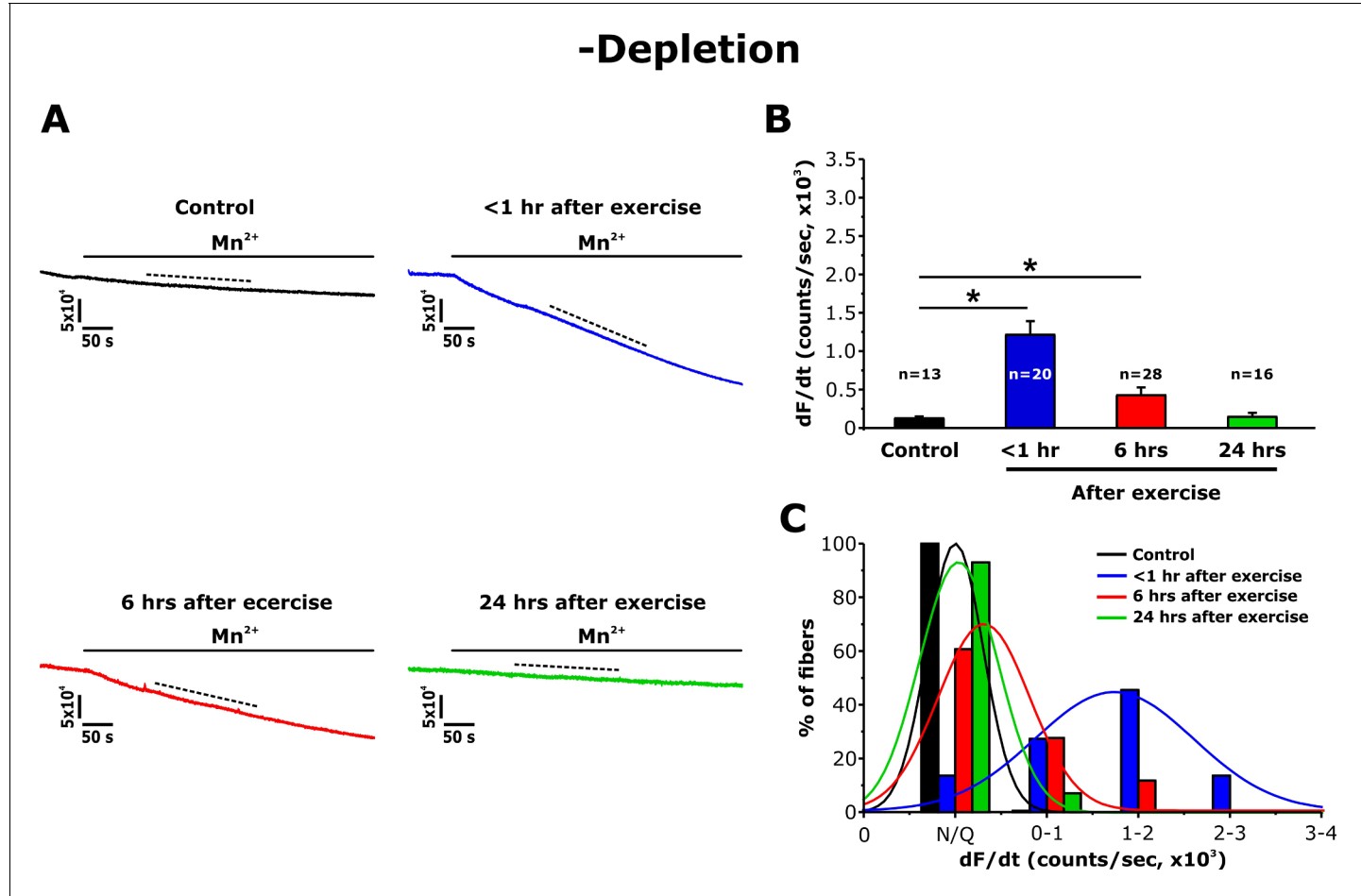

**Figure 4.** Effect of exercise on the maximum rate of $Mn^{2+}$ quench in the absence of store depletion. (**A**) Representative traces of fura-2 fluorescence during application of 0.5 mM $Mn^{2+}$ recorded in FDB fibers in the absence of store depletion (-depletion) isolated from control mice and from mice < 1 hr, 6 hr, and 24 hr after acute treadmill exercise. (**B**) Quantitative analysis of the maximum rate of $Mn^{2+}$ quench in the absence of store depletion. (**C**) Frequency histogram of percentage of fibers exhibiting different levels of maximal rate of $Mn^{2+}$ quench. Histogram data were fit with a single Gaussian distribution. Numbers in bars (n) reflect the number of FDB fibers analyzed. Numbers in bars (n) reflect the number of FDB fibers analyzed. Number of mice used: Control, n = 5;<1 hr after exercise, n = 5; 6 hr after exercise, n = 3; 24 hr after exercise, n = 3; *p<0.05. Data are shown as mean ± SEM.
DOI: https://doi.org/10.7554/eLife.47576.006

The following figure supplement is available for figure 4:

**Figure supplement 1.** Sensitivity of constitutive $Mn^{2+}$ quench in non-depleted FDB fibers after exercise to SOCE inhibitor BTP-2.
DOI: https://doi.org/10.7554/eLife.47576.007

## Peak Ca²⁺ transient amplitude during repetitive stimulation is enhanced <1 hr after exercise

To determine the impact of increased SOCE <1 hr after exercise (*Figures 3* and *4*) on intracellular $Ca^{2+}$ dynamics, myoplasmic and total $Ca^{2+}$ store content were monitored in single FDB fibers isolated from control mice and in mice < 1 hr, 6 hr, and 24 hr after acute treadmill exercise (*Figure 5*). We first assessed the ability of FDB fibers to maintain myoplasmic $Ca^{2+}$ transient amplitude during repetitive high-frequency stimulation (500 ms, 50 Hz, every 2.5 s), using mag-fluo-4, a rapid, low-affinity $Ca^{2+}$ dye to maximize resolution of $Ca^{2+}$ transient magnitude and kinetics (*Capote et al., 2005*). Representative traces showed no significant difference between the four groups in peak $Ca^{2+}$ transient amplitude elicited during the 1st stimulus train (*Figure 5A*). In FDB fibers from control mice, both absolute (*Figure 5B*) and relative (*Figure 5C*) peak $Ca^{2+}$ transient amplitude exhibited a steady modest decline from the first to last stimulation train, with an average decrease of ~30% over

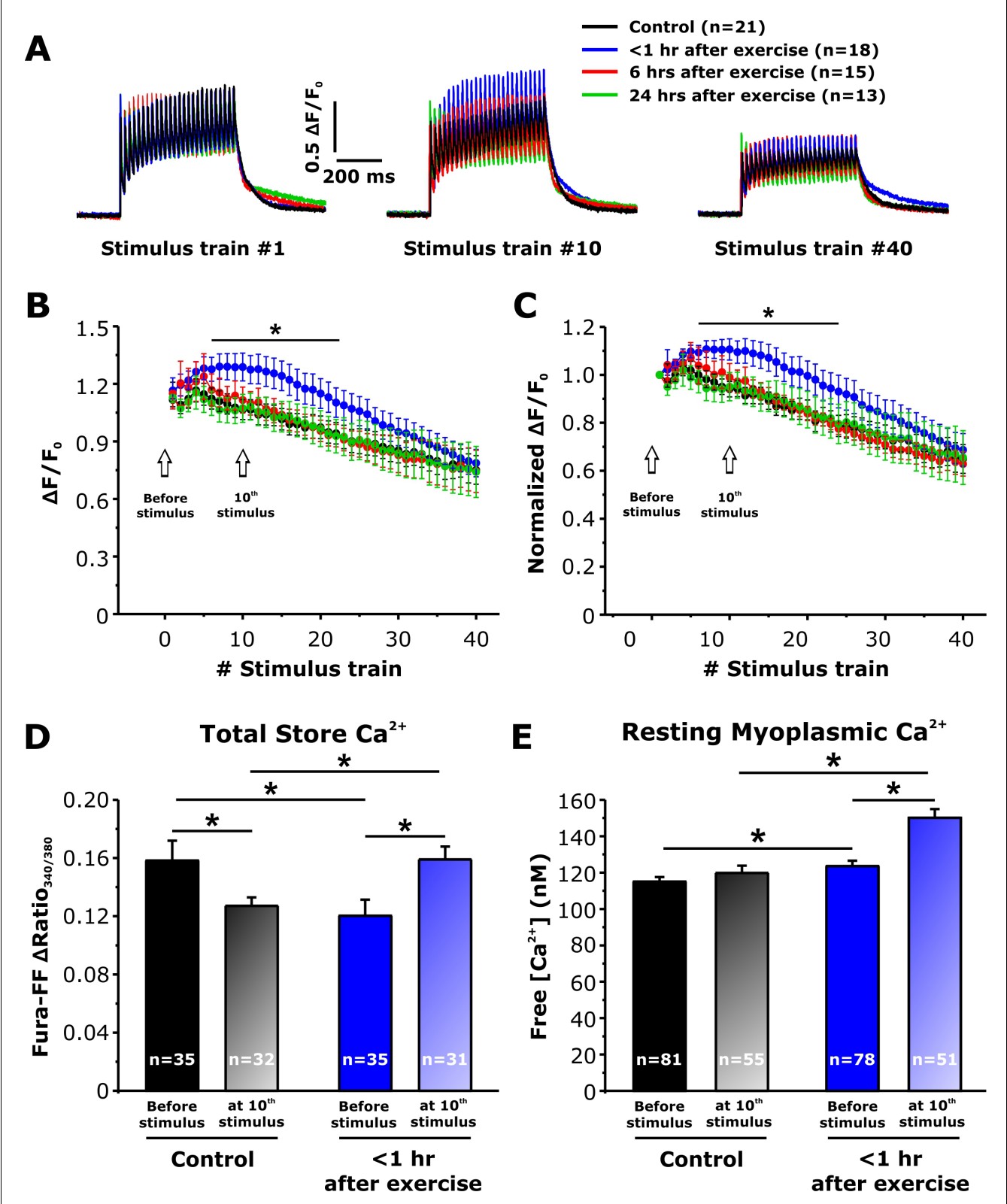

**Figure 5.** Effect of exercise on peak $Ca^{2+}$ transient amplitude during repetitive, high-frequency stimulation. (**A**) Representative superimposed relative ($\Delta F/F_0$) mag-fluo-4 traces during the $1^{st}$, $10^{th}$, and $40^{th}$ stimulation train (500 ms, 50 Hz, every 2.5 s, duty cycle 0.2) in FDB fibers isolated from control mice and from mice < 1 hr, 6 hr, and 24 hr after acute treadmill exercise. (**B and C**) Quantitative analysis of the time course of relative (**B**) and normalized (**C**) change in peak mag-fluo-4 fluorescence. (**D and E**) Effect of high-frequency stimulation on total releasable $Ca^{2+}$ store content in fura-FF-

*Figure 5 continued*

loaded fibers (D) and resting myoplasmicCa$^{2+}$ concentration in fura-2-loaded fibers (E) from control mice and mice < 1 hr after exercise in the absence (solid bars) and following (shaded bars) delivery of 10 stimulus trains (500 ms, 50 Hz, every 2.5 s). n = number of FDB fibers analyzed. Number of mice used in A-C: Control, n = 5;<1 hr after exercise, n = 5; 6 hr after exercise, n = 4; 24 hr after exercise, n = 3. Number of mice used in D: Control, n = 4 (before stimulation) and n = 4 (after stimulation);<1 hr after exercise, n = 3 (before stimulation) and n = 4 (after stimulation). Number of mice used in E: Control, n = 5 (before stimulation) and n = 4 (after stimulation);<1 hr after exercise, n = 5 (before stimulation) and n = 4 (after stimulation); *p<0.05. Data are shown as mean ± SEM.

DOI: https://doi.org/10.7554/eLife.47576.008

The following figure supplements are available for figure 5:

**Figure supplement 1.** Effect of exercise on total Ca$^{2+}$store content and resting myoplasmic Ca$^{2+}$ concentration.

DOI: https://doi.org/10.7554/eLife.47576.009

**Figure supplement 2.** Measurements of free SR Ca levels in D1ER-expressing FDB fibers.

DOI: https://doi.org/10.7554/eLife.47576.010

**Figure supplement 3.** Effect of repetitive, high-frequency stimulation on Ca transient decay kinetics in FDB fibers from control and exercised mice.

DOI: https://doi.org/10.7554/eLife.47576.011

the 40 stimuli. On the other hand, fibers isolated from mice < 1 hr after exercise displayed a significant increase in both absolute and relative peak Ca$^{2+}$ transient amplitude for all trains between stimuli 7–21 (*Figure 5B and C*). Finally, peak Ca$^{2+}$ transient amplitude elicited in fibers isolated from mice either 6 or 24 hr after exercise, were not significantly different from control.

## Total releasable Ca$^{2+}$ Store Content is reduced <1 hr after Exercise, but Increased Following Repetitive, High-frequency Stimulation

The observed increase in peak Ca$^{2+}$ transient amplitude (*Figure 5B*) during repetitive, high-frequency stimulation in fibers from mice < 1 hr after exercise could result from increased SOCE during each stimulus train progressively enhancing total Ca$^{2+}$ store content such that the total releasable Ca$^{2+}$ load is increased. To test this idea, we measured total Ca$^{2+}$ store content in FDB fibers under resting conditions and after delivery of 10 consecutive, high-frequency stimulus trains. Total Ca$^{2+}$ store content was assessed in fura-FF-loaded fibers by application of a Ca$^{2+}$ store release cocktail consisting of 10 µM ionomycin, 30 µM cyclopiazonic acid, and 100 µM EGTA in a Ca$^{2+}$-free Ringer's solution (ICE). The results indicate that compared to fibers from control mice, resting total Ca$^{2+}$ store content was significantly reduced (~25%) in fibers obtained from mice < 1 hr after exercise (*Figure 5D*), but not in fibers isolated from mice 6 and 24 hr after exercise (*Figure 5—figure supplement 1*). Consistent with the increase in constitutive SOCE observed <1 hr after exercise (*Figure 4B*), a significant increase in resting myoplasmic Ca$^{2+}$ concentration was also observed only in fibers < 1 hr after exercise (*Figure 5E* and *Figure 5—figure supplement 1*). Importantly, both total releasable Ca$^{2+}$ store content and myoplasmic resting Ca$^{2+}$ concentration were increased significantly after 10 stimulus trains in fibers from mice < 1 hr after exercise compared to that observed for fibers from both control mice and mice < 1 hr after exercise in the absence of stimulation (*Figure 5D and E*). In contrast, resting free Ca$^{2+}$ concentration in the SR lumen (*Figure 5—figure supplement 2*) was not altered in FDB fibers from mice < 1 hr after exercise.

In addition to an increase in total releasable Ca$^{2+}$ store content, the increase in Ca$^{2+}$ transient amplitude following repetitive, high-frequency stimulation could also result from a reduction in myoplasmic Ca$^{2+}$ buffering and/or SERCA-mediated Ca$^{2+}$ reuptake. To address these possibilities, we compared the kinetics of electrically-evoked twitch Ca$^{2+}$ transient decay in mag-fluo-4-loaded fibers from sedentary mice and mice < 1 hr after exercise before and after delivery of 10 consecutive stimulus trains (500 ms, 50 Hz, every 2.5 s). Using this approach, the decay phase of the electrically-evoked transient is well-described by a double exponential fit in which the fast component primarily reflects Ca$^{2+}$ binding to fast myoplasmic Ca$^{2+}$ buffers while the slow component is dominated by SERCA-mediated SR Ca$^{2+}$ reuptake (*Capote et al., 2005*; *Carroll et al., 1999*; *Baylor and Hollingworth, 2003*) (*Figure 5—figure supplement 3A*). No difference in either the amplitudes or time constants of twitch Ca$^{2+}$ transient decay were observed after delivery of 10 consecutive, high-frequency stimulus trains (*Figure 5—figure supplement 3B–D*) in FDB fibers isolated from either sedentary mice or <1 hr after exercise.

## Maintenance of specific force during repetitive stimulation is increased <1 hr after exercise

We previously reported that sustained EDL contractility during repetitive, high frequency stimulation is increased <1 hr after acute exercise (*Boncompagni et al., 2017*). To determine if the same time course of both enhanced SOCE and sustained $Ca^{2+}$ release correlate with this increment of contractility (*Figures 3–5*), we compared the ability of EDL muscles to maintain force during repetitive stimulation (*Figure 6A*). Average force-frequency curves and peak tetanic-specific force were not significantly different across all four conditions (*Figure 6—figure supplement 1*). All experimental conditions exhibited a similar modest reduction in peak force during the 2nd stimulus train, followed by a rebound increase (referred to as the *bump-phase*) that generally lasted ~10–12 stimulus trains (*Figure 6B and C*). While the peak-specific force generated during the first stimulus train was not different between the four conditions (*Figure 6A*, *left*), maximal specific force increased significantly during stimulus trains 7–30 in EDL muscles from mice < 1 hr after exercise (*Figure 6A*, *middle*), a potentiation that was lost by the end of the protocol (*Figure 6A*, *right*). As a result, both peak-specific force (*Figure 6D*) and the maximum magnitude of the *bump-phase* (*Figure 6E*) were increased 20–25%.

## Orai1 is required for increased SOCE activity and EDL force decay following acute exercise

To address the role of Orai1 channels in the muscle adaptations to exercise reported in *Figures 3–6*, we determined the effect of exercise on SOCE in single FDB fibers and in intact EDL muscles from inducible, muscle-specific Orai1 knockout (iOrai1 KO) and muscle-specific (*Carrell et al., 2016*), dominant-negative Orai1 (dnOrai1) transgenic mice (*Wei-Lapierre et al., 2013*). Consistent with that reported previously (*Carrell et al., 2016*), SOCE induced by pharmacological SR $Ca^{2+}$ depletion was absent in FDB fibers isolated from iOrai1 KO mice (*Figure 7A and B*). Similarly, FDB fibers from iOrai1 KO mice < 1 hr after exercise also lacked detectable $Mn^{2+}$ quench following pharmacological store depletion (*Figure 7A and B*). In addition, constitutive SOCE in the absence of store depletion (-depletion), was also absent in fibers isolated from both control iOrai1 KO mice and iOrai1 KO mice < 1 hr after exercise (*Figure 7C and D*). Finally, the *bump-phase* of muscle contraction during repetitive stimulation was suppressed in EDL muscles from control iOrai1 KO mice and was not further enhanced <1 hr after exercise (*Figure 7E and F*). Similar results were obtained in dnOrai1 mice (*Figure 7—figure supplement 1*), which also specifically lack SOCE in skeletal muscle (*Wei-Lapierre et al., 2013*).

## Discussion

We previously reported that acute treadmill exercise drives the formation of contacts between remodeled SR stacks and extensions of the TT at the I band that promote co-localization of STIM1 and Orai1 proteins. The assembly of these new junctions (CEUs) correlates with reduced EDL muscle force decline during repetitive stimulation (*Boncompagni et al., 2017*). Here, we determined the time course of the disassembly of CEUs following exercise, as well as the impact of these junctions on Orai1-dependent $Ca^{2+}$ entry, myoplasmic $Ca^{2+}$ dynamics, and muscle contractile function. The results, summarized in the model shown in *Figure 8*, demonstrate that: a) CEUs are dynamic structures that assemble during exercise and disassemble following recovery (*Figures 1* and *2*); b) TT retraction from the I band following exercise occurs before SR-stack disassembly (*Figures 1* and *2*); c) peak SOCE activity (i.e. maximum rate of $Mn^{2+}$ quench), reduced $Ca^{2+}$ store content and increased resting $Ca^{2+}$, as well as sustained $Ca^{2+}$ transient amplitude and force production during repetitive stimulation all increase/decrease in parallel with the time course of TT extension within the I band (*Figures 3*, *5* and *6*); d) exercise-induced CEU formation results in constitutively-activated SOCE (*Figure 4*); and e) increased SOCE activity and sustained force production during repetitive stimulation requires Orai1 channels (*Figure 7*). Together, these results indicate that the stacks of flat-parallel SR cisternae formed after exercise are functionally competent CEUs only when associated with a TT extended from the triad.

Two main findings of this study are: a) TTs retract from the CEU following exercise prior to SR-stack disassembly (*Figures 1* and *2*) and b) exercise-dependent increases in Orai1-mediated SOCE

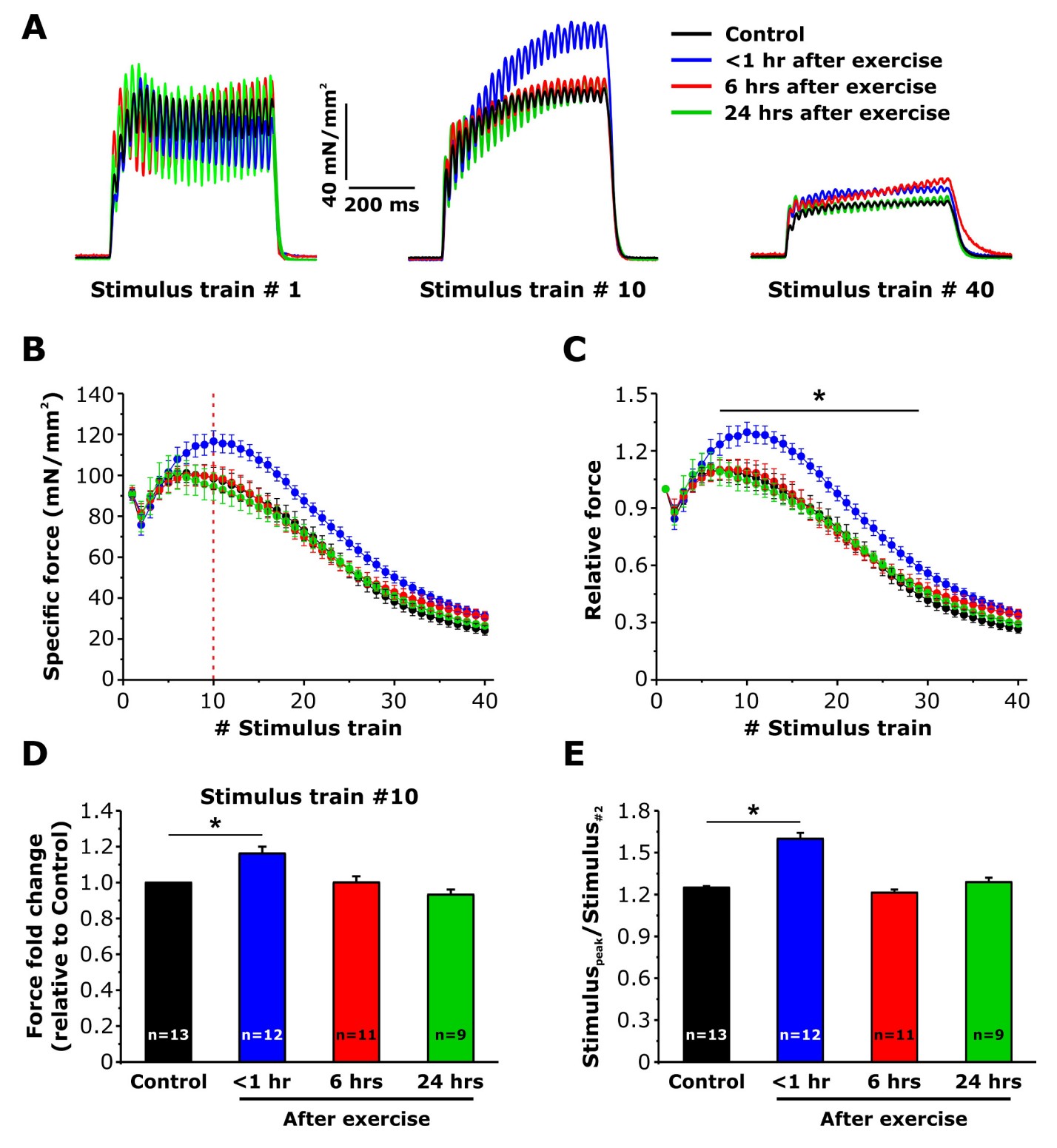

**Figure 6.** Effect of exercise on contractile force during repetitive, high-frequency stimulation. (**A**) Representative specific force traces elicited during the 1st, 10th, and 40th stimulus train (500 ms, 50 Hz, every 2.5 s, duty cycle 0.2) in EDL muscles excised from control mice and from mice < 1 hr, 6 hr, and 24 hr after acute treadmill exercise. (**B and C**) Quantitative analysis of the time course of specific (**B**) and normalized (**C**) peak force during 40 consecutive, high-frequency stimulus trains. (**D**) Quantitative analysis of the fold change of peak-specific force calculated at the 10th stimulus train. (**E**) Quantitative analysis of the ratio between the maximum peak-specific force produced during the 40 stimulus trains and that of the 2nd stimulus train. Numbers in

*Figure 6 continued on next page*

*Figure 6 continued*

bars (n) reflect the number of EDL muscles analyzed. Number of mice used: Control, n = 7;<1 hr after exercise, n = 5; 6 hr after exercise, n = 6; 24 hr after exercise, n = 5; *p<0.05. Data are shown as mean ± SEM.

DOI: https://doi.org/10.7554/eLife.47576.012

The following figure supplement is available for figure 6:

**Figure supplement 1.** Effect of exercise on EDL muscle force-frequency curve and maximum tetanic force.

DOI: https://doi.org/10.7554/eLife.47576.013

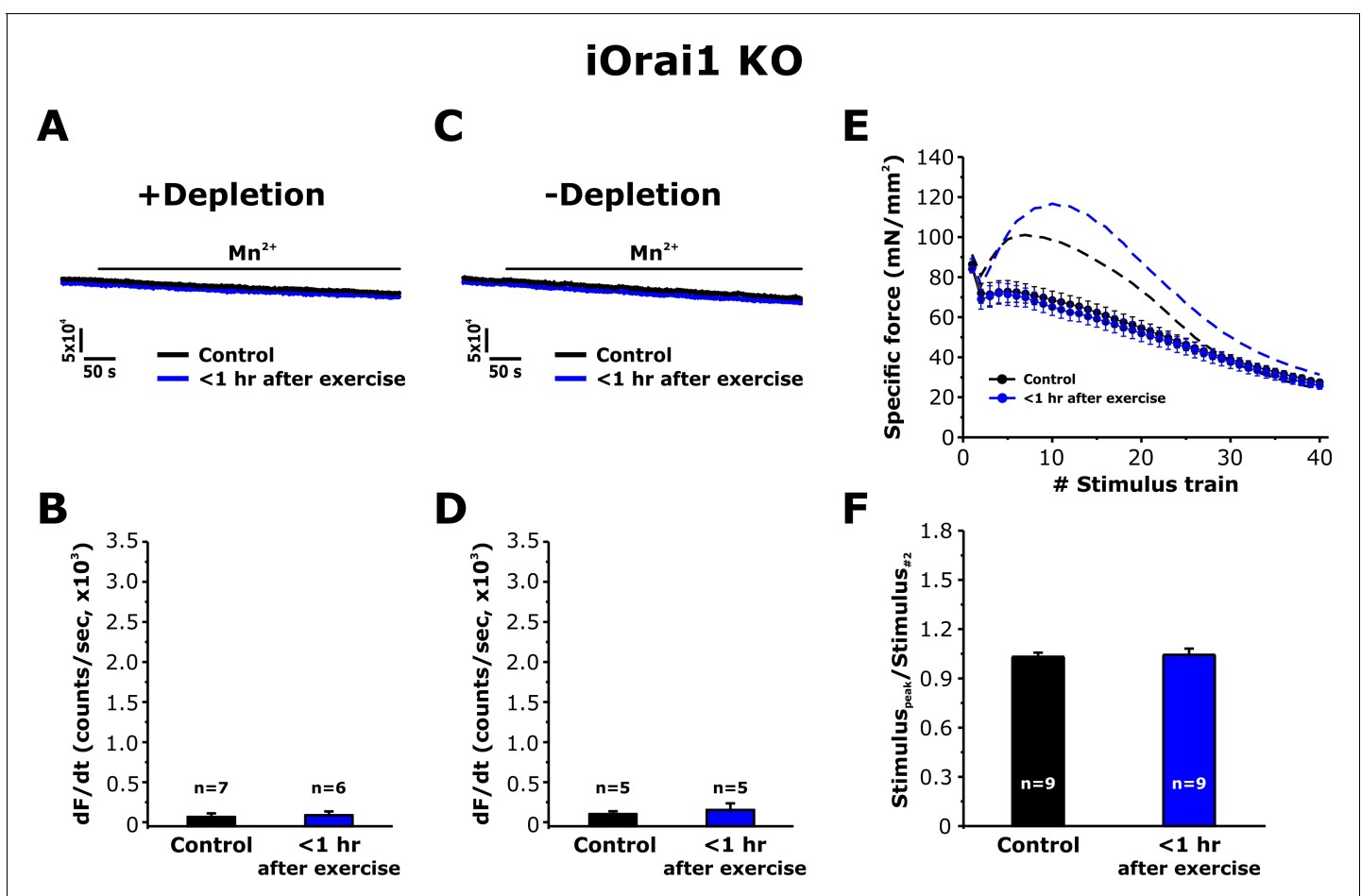

**Figure 7.** Effect of exercise on the maximum rate of $Mn^{2+}$ quench and contractile force in iOrai1 KO mice. Data were obtained in EDL muscles and FDB fibers from non-exercised iOrai1 KO mice and iOrai1 KO mice < 1 hr after acute treadmill. (**A**) Representative traces of fura-2 fluorescence during application of 0.5 mM $Mn^{2+}$ recorded in FDB fibers following store depletion with 1 μM thapsigargin + 15 μM cyclopiazonic (+depletion). (**B**) Representative traces of fura-2 fluorescence during application of 0.5 mM $Mn^{2+}$ recorded in FDB the absence of store depletion (-depletion). (**C–D**) Quantitative analysis of the maximum rate of $Mn^{2+}$ quench under store-depleted (**C**) and non-depleted (**D**) conditions in FDB fibers. (**E**) Quantitative analysis of the time course of specific force during 40 consecutive, high-frequency stimulus trains in EDL muscles. (**F**) Quantitative analysis of the ratio between the maximum peak-specific force produced during the 40 stimulus trains and that of the 2nd stimulus train in EDL muscles. Numbers (n) reflect the number of FDB fibers or muscles analyzed. Number of mice used: iOrai1 KO control, n = 5; iOrai1 KO exercised, n = 5; *p<0.05. Data are shown as mean ± SEM.

DOI: https://doi.org/10.7554/eLife.47576.014

The following figure supplement is available for figure 7:

**Figure supplement 1.** Effect of exercise on the maximum rate of $Mn^{2+}$ quench and contractile force in dnOrai1 mice.

DOI: https://doi.org/10.7554/eLife.47576.015

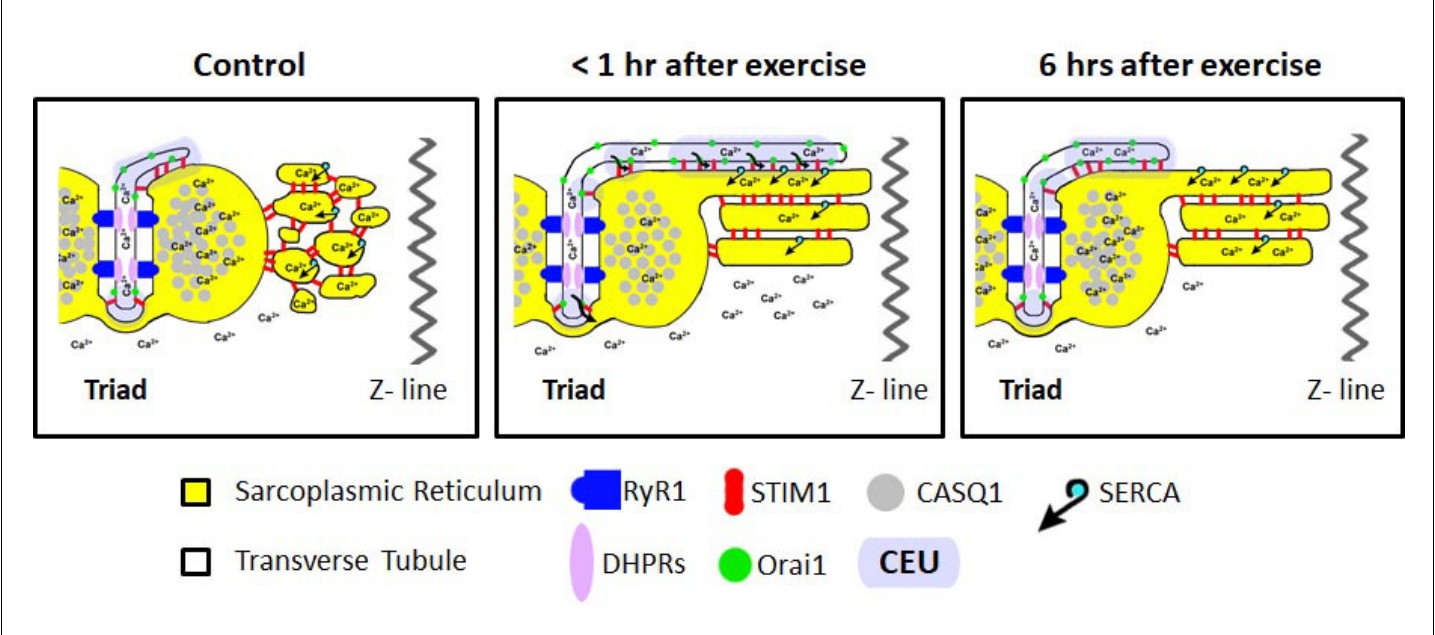

**Figure 8.** Model of structural events and functional effects of CEU assembly and disassembly following acute exercise and during recovery. Left) In non-exercised or sedentary muscle, the SR is fully replete with $Ca^{2+}$ ions, and the myoplasmic $Ca^{2+}$ concentration is low. STIM1 proteins are located throughout the I band, while Orai1 proteins are almost exclusively within the TT membrane. Middle) Less than 1 hr after acute exercise, total $Ca^{2+}$ store content is reduced, and myoplasmic $Ca^{2+}$ is modestly elevated. In addition, the free-SR undergoes a remodeling that results in the formation of SR-stacks, while the TT elongates from the triad into the I band to create junctional contacts, referred to as *Calcium Entry Units* (CEUs). Right) 6 hr of recovery after acute exercise, the SR is fully replete with $Ca^{2+}$, the myoplasmic $Ca^{2+}$ concentration returns to control levels, and the TT extension is largely retracted back from contacts with SR-stacks.
DOI: https://doi.org/10.7554/eLife.47576.016

(*Figures 3*, *4* and *7*), intracellular $Ca^{2+}$ dynamics (*Figure 5*), and maintained contractile activity during repetitive stimulation (*Figure 6*) declines with a time course like that of TT retraction. Together, these results show that the functional effects of acute exercise are controlled by association of TTs (containing Orai1 proteins) with SR-stacks (containing STIM1 proteins) in the I band, with an increase of TT/SR contact length that provides a greater surface area for STIM1-Orai1 interaction. The role of SR-SR interfaces that lack a TT element remains unclear. However, SR-stacks containing multiple elements virtually identical to those formed in muscle after exercise, are also observed in non-muscle cells following STIM1 overexpression (*Orci et al., 2009*; *Perni et al., 2015*). Interestingly, both ER-ER contacts in non-muscle cells overexpressing STIM1 and SR-SR contacts in muscle after acute exercise exhibit small electron dense bridges within a similar junctional gap width (~8 nm). These similarities in membrane structure and gap width are consistent with the small electron dense bridges observed in EM images reflecting STIM1 proteins. Similar to the 'diffusion trap' model of STIM1-Orai1 coupling in non-excitable cells proposed previously (*Wu et al., 2014*), one possible interpretation of these findings is that STIM1-containing SR-stacks could be needed to efficiently trap Orai1 channels during TT elongation into the I band.

An unexpected finding of this study was the demonstration of significant $Mn^{2+}$ quench in the absence of thapsigargin/cyclopiazonic acid-induced store depletion (constitutive $Ca^{2+}$ entry) in FDB fibers isolated from mice < 1 hr after treadmill exercise (*Figure 4*). As the constitutive $Ca^{2+}$ entry in these fibers was rapidly blocked by application of BTP-2 (*Figure 4—figure supplement 1*), the constitutive entry in these fibers is unlikely to represent increased susceptibility of fibers to damage shortly after exercise. In support of this idea, the constitutive $Ca^{2+}$ entry observed <1 hr after exercise is absent in fibers from two separate mouse models deficient in Orai1 function (*Figure 7* and *Figure 7—figure supplement 1*). These findings suggest that the constitutive $Ca^{2+}$ entry is mediated by Orai1 channels (rather than TrpC or mechano-sensitive channels). D1ER measurements of 'spatially-averaged' free $Ca^{2+}$ concentration in the SR were not significantly different after exercise

(though areas of local depletion are possible). It is important to note, however, that D1ER localization studies indicate that the majority of the D1ER signal originated from the triad junction (*Figure 5—figure supplement 2A*). Thus, the D1ER measurements suggest that the free $Ca^{2+}$ concentration within the triad junction is not different after exercise. Thus, constitutive $Ca^{2+}$ entry observed after exercise most likely does not originate from Orai1 channels within the triad junction. Rather, constitutive entry may arise from Orai1 channels in another location, possibly CEUs assembled during exercise in the I band region of the sarcomere that are not well-reflected in the spatially-averaged D1ER signal. These observations support the possibility that CEUs represent the site of constitutive $Ca^{2+}$ entry after exercise. An alternative explanation is that exercise could produce a signal that directly activates Orai1 independent of store depletion.

Assuming no changes in the properties of sarcolemmal $Ca^{2+}$ efflux mechanisms, the increase in constitutive $Ca^{2+}$ entry observed <1 hr after exercise could explain the elevation in resting $Ca^{2+}$ concentration observed at this time point (*Figure 5E* and *Figure 5—figure supplement 1*). Again, partial $Ca^{2+}$ depletion within SR-stacks of the CEU represents one possible mechanism for the constitutive entry observed immediately following exercise. Alternatively, as lower levels of store depletion are sufficient for Stim2 to activate Orai1-dependent $Ca^{2+}$ entry (*Brandman et al., 2007*; *Thiel et al., 2013*), the constitutive $Ca^{2+}$ entry observed after exercise could reflect Stim2-mediated activation of Orai1 channels within the CEUs. Thus, it will be important for future studies to more definitively assess SERCA and Stim2 expression within CEUs, determine the relative distribution of the short and long forms of STIM1 within CEUs (*Michelucci et al., 2018*), and quantitatively model $Ca^{2+}$ binding and diffusion within and out of these structures.

Another observation that deserves consideration is the non-linear correspondence between exercise-induced changes in structure (TT/SR contact length increases five fold; *Figure 2E*) and SOCE function (~65% increase in maximal rate of $Mn^{2+}$ quench; *Figure 3B*) in FDB muscle. One possibility is that, as proposed above, CEUs may reflect the subcellular location for the constitutive SOCE observed <1 hr after exercise. However, given that total releasable $Ca^{2+}$ store content at rest was reduced <1 hr after exercise, it is possible that pre-existing STIM and Orai1 complexes distinct from newly formed CEUs could also contribute to the observed increase in SOCE function.

We investigated the mechanism by which increased SOCE following acute exercise increases peak $Ca^{2+}$ transient amplitude during repetitive, high-frequency stimulation. Conceivably, increased total (*Figure 3*) and constitutive (*Figure 4*) SOCE activity <1 hr after exercise could potentiate peak $Ca^{2+}$ transient amplitude either by increasing total releasable SR $Ca^{2+}$ content or elevating myoplasmic $Ca^{2+}$ levels enough to reduce fast buffering. Alternatively, SERCA activity could be inhibited after exercise due to glycogen depletion. Kinetic analysis of twitch $Ca^{2+}$ transient decay before and after 10 stimulus trains in FDB fibers from control and exercised mice did not reveal a significant change in either myoplasmic $Ca^{2+}$ buffering or SERCA-mediated SR $Ca^{2+}$ reuptake (*Figure 5—figure supplement 3*). On the other hand, both total releasable $Ca^{2+}$ store content (*Figure 5D*) and resting myoplasmic $Ca^{2+}$ concentration (*Figure 5E*) were significantly increased after 10 stimulus trains in FDB fibers from mice < 1 hr after exercise. Together, these findings are consistent with the observed increase in peak $Ca^{2+}$ transient amplitude (and force production) after acute exercise resulting, at least in part, from enhanced SOCE activity promoting SR refilling and total releasable $Ca^{2+}$ content to a level sufficient to augment $Ca^{2+}$ flux.

EDL muscles isolated from mice < 1 hr after treadmill exercise exhibit an increased ability to produce sustained force during repetitive stimulation (*Figure 6*) (*Boncompagni et al., 2017*). Muscle force production shows a rebound increment in peak-specific force (between stimulus trains 2–12), referred to as the *bump-phase* of the repetitive stimulation contraction curve. The following observations are consistent with this *bump-phase* reflecting the contribution of Orai1-dependent SOCE: 1) it is reduced by interventions (BTP-2, 2-APB, 0 $Ca^{2+}$) that reduce SOCE (*Boncompagni et al., 2017*); 2) it is increased <1 hr after treadmill exercise when SOCE is also increased; 3) it is unaltered 6 and 24 hr after treadmill exercise when SOCE activity has returned to control levels; 4) it is absent in mice lacking Orai1-dependent $Ca^{2+}$ entry both under control conditions and after acute exercise. As some mouse models have been shown to exhibit an increase in basal levels of CEUs similar to those observed immediately following exercise (*Boncompagni et al., 2012a*; *Boncompagni et al., 2012b*; *Ko et al., 2011*), it will be important for future studies to determine if muscles from these mice also exhibit an increase in both SOCE and the bump-phase of the repetitive stimulation contraction curve.

In conclusion, the findings reported here support the idea that CEUs formed after acute exercise represent an adaptive response of skeletal muscle to exercise that augments Orai1-dependent $Ca^{2+}$ entry to optimize myoplasmic $Ca^{2+}$ dynamics in a manner that limits muscle force decline during sustained activation.

## Materials and methods

### Animals

Male wild type (WT) C57Bl/6 mice (4–6 months old) were housed in microisolator cages at 20°C on a 12 hr light/dark cycle while being provided free access to standard chow and water. Two previously described muscle-specific mouse lines deficient in Orai1 function were also employed: a) dominant-negative Orai1 (dnOrai1) transgenic mice (*Wei-Lapierre et al., 2013*) and b) tamoxifen-inducible, Orai1 (*Orai1fl/fl::ACTAMer-Cre* or iOrai1) knockout mice (*Carrell et al., 2016*). Four-month old male WT, dnOrai1, and iOrai1 KO mice were randomly assigned to two experimental groups: control and exercised (mice exposed to a single bout of treadmill exercise; see below). All animal studies were designed to minimize animal suffering and were approved by the Italian Ministry of Health (992/2019-PR) and the University Committee on Animal Resources at the University of Rochester (UCAR-2006-114E).

### Tamoxifen treatment

Two-three-month-old *Orai1fl/fl::ACTAMer-Cre* mice were provided ad libitum access to tamoxifen-infused mouse chow (Envigo, Huntingdon, UK) for a period of 4 weeks, and then returned to regular chow for at least one additional week before being used for experiments. Semi-quantitative PCR of Orai1 transcript level was used to verify success of tamoxifen treatment in Orai1 ablation in muscle, as previously described (*Carrell et al., 2016*).

### Treadmill exercise protocol

A standardized acute exercise protocol was performed at room temperature using a running treadmill (Columbus Instruments, Columbus, OH, USA) on a flat surface (0° incline) as previously described (*Boncompagni et al., 2017*). Briefly, immediately after a warm-up period (10 min at 5 m/min), mice were subjected to a 65 min exercise protocol consisting of an initial 25 min at a speed of 10 m/min, followed by 20 min at 15/m/min, 15 min at 20 m/min and then five final 1 min intervals where the speed was increased an additional 1 m/min for each interval. The protocol was stopped when mice either reached the end of the protocol or were unable to continue as indicated by the inability of the animal to maintain running and contact with the treadmill belt (see 10 for details). Following treadmill exercise, EDL muscles (for ex vivo muscle contractility studies) and FDB muscles (for $Mn^{2+}$ quench/$Ca^{2+}$ measurements) were removed and prepared for experiments at one of the following three time points: 1)<1 hr from the end of the in vivo exercise protocol (referred to as <1 hr after exercise), 2) 6 hr after exercise, or 3) 24 hr after exercise.

### Electron microscopy (EM)

Intact EDL and FDB muscles were fixed at room temperature with 3.5% or 6% glutaraldehyde in 0.1M sodium cacodilate (NaCaCO) buffer (pH 7.2), and processed for EM acquisition as previously described (*Boncompagni et al., 2009*). For TT staining, specimens were post-fixed in a mixture of 2% $OsO_4$ and 0.8% $K_3Fe(CN)_6$ for 1–2 hr followed by a rinse with 0.1M NaCaCO buffer with 75 mM $CaCl_2$. Ultrathin sections (~50 nm) were cut using a Leica Ultracut R microtome (Leica Microsystems, Vienna, Austria) with a Diatome diamond knife (Diatome, Biel, Switzerland) and double-stained with uranyl acetate and lead citrate. Sections were viewed in a FP 505 Morgagni Series 268D electron microscope (FEI Company, Brno, Czech Republic), equipped with Megaview III digital camera (Olympus Soft Imaging Solutions, Munster, Germany) and Soft Imaging System at 60 kV (100 kV for TT staining).

### Quantitative analyses of EM images

The percentage of fibers exhibiting SR-stacks and the number of SR-stacks per 100 μm² were determined from electron micrographs of non-overlapping regions randomly collected from transversal

EM sections, as described previously (*Boncompagni et al., 2017*). For each specimen, 10–15 representative fibers were analyzed and 5 micrographs at 28000x magnification were taken for each fiber. Operationally, 'stacks' were defined as two or more adjacent SR flat elements of stackable cisternae separated by a junctional gap width of ~8 nm (*Boncompagni et al., 2017*). The average length of SR-stacks was measured in micrographs taken at 44,000x magnification. TT/SR contact length (i.e. length of TT adjacent to SR stack membrane) and extensions of the TT network within the I band of sarcomere (total TT length) were measured in electron micrographs of non-overlapping regions randomly collected from transversal EM sections. For each specimen, 10–15 representative fibers were analyzed and 5 micrographs at 28,000x magnification in transverse sections of samples stained with ferrocyanide were taken for each fiber. Quantitative analyses of SR-stacks, TT/SR contact length and total TT length were obtained using the Analy-SIS software of the EM digital camera (Olympus Soft Imaging Solutions, Munster, Germany).

## Isolation of single FDB muscle fibers

As healthy single fibers are readily isolated from FDB muscles, all $Mn^{2+}$ quench and $Ca^{2+}$ measurements were conducted in single, acutely dissociated FDB fibers. FDB muscles were dissected from hind paws and placed in a dish containing Ringer's solution consisting of (in mM): 145 NaCl, 5 KCl, 2 $CaCl_2$, 1 $MgCl_2$, and 10 HEPES, pH 7.4. Muscles were then incubated in Ringer's solution supplemented with 1 mg/ml collagenase A (Roche Diagnostics, Indianapolis, IN, USA) for 60 min while rocking gently at 37°C to allow enzymatic dissociation. Single FDB fibers obtained by mechanical trituration were plated on glass-bottom dishes and allowed to settle for >20 min before experimentation (see below). Only fibers with clear striations and no signs damage were used for recordings.

## $Mn^{2+}$ quench measurements

FDB fibers were loaded with 5 μM fura-2 AM for 1 hr at 37° C in a $Ca^{2+}$-free Ringer's solution containing (in mM): 145 NaCl, 5 KCl, 1 $MgCl_2$, 0.2 EGTA, pH 7.4. To measure the maximal rate of $Mn^{2+}$ quench, during fura-2 AM loading, fibers were also incubated with two SERCA pump inhibitors (1 μM thapsigargin; 15 μM cyclopiazonic acid) to fully deplete SR $Ca^{2+}$ stores (+depletion) and a skeletal muscle myosin inhibitor (30 μM *N*-benzyl-p-toluene sulfonamide [BTS]) to prevent movement artifacts (*Wei-Lapierre et al., 2013*). B) In a second set of studies, FDB fibers were loaded with fura-2 AM and BTS in the absence of SERCA pump inhibitors (-depletion). Both store-depleted and non-depleted FDB fibers were then bathed in $Ca^{2+}$-free Ringer's and excited at 362 nm (isobestic point of fura-2), while emission was detected at 510 nm using a DeltaRam illumination system (Photon Technologies Inc, Birmingham, NJ, USA). After obtaining an initial basal rate of fura-2 decay ($R_{baseline}$), fibers were exposed to $Ca^{2+}$-free Ringer's supplemented with 0.5 mM $MnCl_2$. The maximum rate of fura-2 quench in the presence of $Mn^{2+}$ ($R_{max}$) was then obtained from the peak differential of the fura-2 emission trace during $Mn^{2+}$ application. The maximum rate of SOCE ($R_{SOCE}$) was calculated as $R_{SOCE} = R_{max} R_{baseline}$ and expressed as dF/dt in counts/sec (*Wei-Lapierre et al., 2013*).

## $Ca^{2+}$ transient measurements

Myoplasmic $Ca^{2+}$ transients were monitored in acutely isolated FDB fibers as described previously (*Ainbinder et al., 2015*). Briefly, FDB fibers were loaded with 4 μM mag-fluo-4-AM for 20 min at room temperature followed by washout in dye-free solution supplemented with 25 μM BTS for 20 min. While continuously perfused with a control Ringer's solution supplemented with 25 μM BTS, fibers were electrically stimulated with a repetitive stimulation protocol (40 consecutive, 500 ms duration, 50 Hz stimulus trains delivered every 2.5 s) using an extracellular electrode placed adjacent to the fiber. Mag-fluo-4 was excited at 480 ± 15 nm using an Excite epifluorescence illumination system (Nikon Instruments, Melville, NY, USA) and fluorescence emission at 535 ± 30 nm was monitored with a 40X oil objective and a photomultiplier detection system (Photon Technologies Inc, Birmingham, NJ, USA). Relative changes in mag-fluo-4 fluorescence from baseline ($F/F_0$) were recorded at 10 kHz and analyzed using Clampfit 10.0 (Molecular Devices, Sunnyvale, CA, USA).

The decay kinetics of single electrically-evoked twitch $Ca^{2+}$ transients were determined in mag-fluo-4-loaded FDB fibers isolated from both sedentary mice and mice < 1 hr after exercise. For these experiments, 10 electrically-evoked twitch transients (0.5 Hz) were collected both before and after delivery of 10 stimulus trains (500 ms, 50 Hz, every 2.5 s). The decay phase of each transient was

fitted according to the following equation and used to generate average values for both before and after high-frequency stimulation:

$$F(t) = A_{fast} \times [exp(-t/\tau_{fast})] + A_{slow} \times [exp(-t/\tau_{slow})]$$

where F($t$) is the fluorescence at time $t$, $A_{fast}$ and $\tau_{fast}$ are the amplitude and time constants of the fast component, and $A_{slow}$ and $\tau_{slow}$ are the amplitude and time constants of the slow component.

## Total Ca$^{2+}$ store content measurements

Total releasable Ca$^{2+}$ store content was determined in FDB fibers loaded with 4 µM fura-FF AM, a low affinity ratiometric Ca$^{2+}$ dye, for 30 min at room temperature in control Ringer's solution followed by 30 min washout in dye-free Ringer's solution supplemented with 40 µM BTS as described previously (*Kimura et al., 2009*; *Loy et al., 2011*). Briefly, fura-FF-loaded fibers were perfused in Ca$^{2+}$-free Ringer's solution while alternately excited at 340 and 380 nm (510 nm emission) every 250 ms (30 ms exposure per wavelength and 2 × 2 binning) using a monochromator-based illumination system (TILL Photonics, Graefelfing, Germany). Fura-FF emission at 535 ± 30 nm was captured using a high speed, digital QE CCD camera (TILL Photonics, Graefelfing, Germany). Total releasable Ca$^{2+}$ store content was assessed from the difference between basal and peak fura-FF ratio ($\Delta$Ratio$_{340/380}$) upon application of a Ca$^{2+}$ release cocktail containing 10 µM ionomycin, 30 µM CPA, and 100 µM EGTA in a Ca$^{2+}$-free Ringer's solution (ICE). For a subset of experiments, total releasable Ca$^{2+}$ store content was measured in fura-FF-loaded fibers isolated from control mice and mice < 1 hr after treadmill exercise following delivery of 10 consecutive high-frequency stimulus trains (500 ms, 50 Hz, every 2.5 s). To confirm that the peak fura-FF signal during ICE application was not saturated, maximal fura-FF responsiveness was assessed at the end of each experiment by subsequent application of Ca$^{2+}$-containing Ringer's solution. Analysis of peak ICE-induced change in fura-FF ratio ($\Delta$Ratio$_{340/380}$) was calculated using Clampfit 10.0 (Molecular Devices, Sunnyvale, CA, USA).

## Resting Ca$^{2+}$ measurements

Resting myoplasmic Ca$^{2+}$ concentration was determined in FDB fibers loaded with 5 µM fura-2 AM for 30 min at room temperature in control Ringer's solution followed by a 30 min washout with dye-free Ringer's solution. Fura-2-loaded fibers were placed on the stage of an inverted epifluorescence microscope (Nikon Instruments) and alternatively excited at 340 and 380 nm (30 ms exposure per wavelength and 2 × 2 binning) using a monochromator-based illumination system with fluorescence emission at 510 nm was captured using a high speed, digital QE CCD camera (TILL Photonics, Graefelfing, Germany). Fura-2 340/380 ratios from myoplasmic areas of interest were calculated using TILL visION software (TILL Photonics, Graefelfing, Germany), analyzed offline using NIH ImageJ, and converted to resting free Ca$^{2+}$ concentrations using a calibration curve for fura-2 generated as described previously (*Lanner et al., 2012*).

## SR-free Ca$^{2+}$ measurements

The free Ca$^{2+}$ concentration in the SR lumen was assessed in FDB fibers expressing the D1ER cameleon Ca$^{2+}$ sensor. D1ER was expressed in FDB fibers using an in vivo electroporation approach described previously (*Canato et al., 2010*). Briefly hind limb footpads of anesthetized mice were first injected with bovine hyaluronidase (6 µl, 0.4 U/ µl) and then one hour later with D1ER cDNA (20 µg in 71 mM NaCl) using 30-gauge needles. The footpad was then electroporated with 20 stimulations of 100 V/cm, 20 ms duration delivered at 1 Hz using subcutaneous electrodes placed perpendicular to the long axis of the muscle, close to the proximal and distal tendons. Fibers were isolated as described above and used for experiments either 7–14 days later.

D1ER-expressing fibers from control mice and mice < 1 hr after exercise were bathed in control Ringer's solution supplemented with 25 µM BTS on the stage of an inverted Nikon Eclipse TE-2000-U microscope (Nikon Instruments, Melville, NY, USA). A relatively large rectangular region of interest of the fiber was excited at 436 nm (10 nm bandwidth). Fluorescence emission within the region of interest was split with a 515 nm dichroic mirror and collected at 480 nm (30 nm bandwidth) and 535 (30 nm bandwidth) using a photomultiplier counting system (Chroma Technology, Bellows Falls, VT). In order to directly compare measurements across multiple fibers, all experiments used identical excitation and emission gains. Following subtraction of background fluorescence, YFP and CFP emission intensities were collected and digitized at 100 Hz first in the presence of control

calcium-free Ringer's solution supplemented with 40 µM BTS and then during perfusion of ICE supplemented with 40 µM BTS. The ratio of YFP to CFP emission ($R = F_{YFP}/F_{CFP}$) for each fiber was calculated offline with averages calculated both immediately before application of ICE and then again after reaching a minimum value in the presence of ICE.

### Immunofluorescence

FDB fibers dissociated from control mice and mice < 1 hr after exercise were plated on glass bottom dishes and fixed in 4% paraformaldehyde for 20 min at room temperature. Following blocking for 1 hr at room temperature in 10% BSA in 1X PBS-T (PBS + 0.1% Triton X-100), fibers were co-labeled overnight at 4°C using primary antibodies for GFP (Rabbit polyclonal antibody, Thermo Fisher Scientific, Cat. # A11122, 1:3000) and either α-actinin (Mouse anti α-actinin, Sigma, Cat # A7811, 1:750) or the type1 ryanodine receptor (RYR1- Mouse anti RYR1 Cat # 34C, DHSB University of Iowa, 1:30). All primary antibody incubations were in 2% BSA in 1X PBS-T. Following three 10 min washes in PBS-T, samples were incubated with a 1:500 dilution of Goat anti Rabbit Alexa fluor 488 (Molecular Probes Cat # A11034) and Goat Anti Mouse Alexa fluor 594 (Molecular Probes Cat # R37121) for 1 hr at room temperature. Following three 10 min washes in 1X PBS-T, samples were imaged using an Olympus FV1000 laser scanning confocal microscope (Olympus Scientific Solutions, Wlatham, MA) and a 100X, UPlanSAPO NA 1.4 oil immersion objective. Alexa 488 and 594 were sequentially excited at 488 and 559 nm and detected at 515 and 617 nm respectively.

### Ex vivo muscle contractility measurements

Ex vivo assessment of muscle force production was made in intact EDL muscles excised from control mice and from mice < 1 hr, 6 hr or 24 hr after acute treadmill exercise. Mice were anesthetized by intra-peritoneal injection of anesthetic (*Wei-Lapierre et al., 2013*). EDL muscles were isolated, tied using 4–0 surgical suture, excised, and then attached to a servo motor and force transducer (Aurora Scientific, Aurora, Ontario, Canada) between two platinum electrode plates in a chamber continuously perfused with oxygenated Ringer solution containing (in mM): 137 NaCl, 5 KCl, 1.2 NaH$_2$PO$_4$, 1 MgSO$_4$, 2 CaCl$_2$,10 glucose, and 24 NaHCO$_3$. Before starting measurements, optimal stimulation level and muscle length ($L_0$) were determined using a series of 1 Hz-twitch stimulation trains while stretching the muscle to a length that generated maximal force ($F_0$). After establishing $L_0$, muscles were first equilibrated using three tetani (500 ms, 150 Hz) given at 1 min intervals and then subjected to a standard force frequency protocol (from 1 to 250 Hz). After 3 min of rest, a single sustained high frequency tetanus (150 Hz, 2 s). After an additional 5 min of rest, muscles were then subjected to a repetitive stimulation protocol (40 consecutive, 500 ms duration, 50 Hz stimulus trains delivered every 2.5 s). All muscle contractility experiments were carried out at 30°C. Muscle force was recorded using Dynamic Muscle Control software and analyzed using a combination of both Dynamic Muscle Analysis (Aurora Scientific, Aurora, Ontario, Canada) and Clampfit 10.0 (Molecular Devices, Sunnyvale, CA, USA) software. Specific force was calculated by normalizing the absolute force to the physiological cross sectional area as previously described (*Boncompagni et al., 2017*).

### Statistical analyses

Statistical significance was determined using either Student's t-test for comparing two groups (*Figure 7* and S3) and a one-way ANOVA followed by *post-hoc* Tukey test when making multiple comparisons (all other figures). Amplitude histograms of maximal rates of Mn$^{2+}$ quench were fit according to a single Gaussian distribution. In all cases, differences were considered as statistically significant at *$p<0.05$. All summary data were presented as mean ± SEM.

## Acknowledgements

This work was supported by grants from the National Institutes of Health (AR059646 to RTD and subcontract to FP), the Italian MIUR (PRIN #2015ZZR4W3 to FP), and the Alfred and Eleanor Wedd Fund (to AM).

## Additional information

### Funding

| Funder | Grant reference number | Author |
|---|---|---|
| National Institutes of Health | AR059646 | Robert T Dirksen<br>Feliciano Protasi |
| Ministry of Education, University and Research | PRIN #2015ZZR4W3 | Feliciano Protasi |
| Alfred and Eleanor Wedd Fund, University of Rochester | | Antonio Michelucci |

The funders had no role in study design, data collection and interpretation, or the decision to submit the work for publication.

### Author contributions

Antonio Michelucci, Simona Boncompagni, Conceptualization, Data curation, Formal analysis, Validation, Investigation, Visualization, Methodology, Writing—original draft, Writing—review and editing; Laura Pietrangelo, Data curation, Formal analysis, Validation, Investigation, Visualization; Maricela García-Castañeda, Conceptualization, Data curation, Formal analysis, Investigation, Methodology; Takahiro Takano, Sundeep Malik, Conceptualization, Data curation, Formal analysis, Methodology; Robert T Dirksen, Conceptualization, Resources, Data curation, Formal analysis, Supervision, Funding acquisition, Validation, Investigation, Visualization, Methodology, Writing—original draft, Project administration, Writing—review and editing; Feliciano Protasi, Conceptualization, Data curation, Formal analysis, Supervision, Funding acquisition, Validation, Investigation, Visualization, Methodology, Writing—original draft, Project administration, Writing—review and editing

### Author ORCIDs

Antonio Michelucci (iD) https://orcid.org/0000-0002-0720-3009
Simona Boncompagni (iD) http://orcid.org/0000-0001-5308-5069
Robert T Dirksen (iD) https://orcid.org/0000-0002-3182-1755

### Ethics

Animal experimentation: All animal studies were designed to minimize animal suffering and were approved by the Italian Ministry of Health (992/2019-PR) and the University Committee on Animal Resources at the University of Rochester (UCAR-2006-114E).

### Decision letter and Author response

Decision letter https://doi.org/10.7554/eLife.47576.019
Author response https://doi.org/10.7554/eLife.47576.020

## Additional files

### Supplementary files

• Transparent reporting form DOI: https://doi.org/10.7554/eLife.47576.017

### Data availability

All data generated or analysed during this study are included in the manuscript and supporting files.

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
