## [Decision Letter]

**Acceptance summary:**

Following exercise, muscle fibers undergo membrane remodeling that results in increased Ca^2+^ release and muscle twitch strength. Previous work has shown that after exercise the T tubule (TT) invaginations of the plasma membrane elongate and make close appositions to newly formed stacks of sarcoplasmic reticulum (SR), and that the presence of the STIM1 Ca^2+^ sensor and Orai1 calcium channels at these newly formed junctions may increase the loading of calcium into the SR to promote contractile force. Here the authors provide new evidence for this mechanism, showing that after exercise the appearance and disappearance of new SR stack-TT structures parallels the time course of augmented calcium entry through Orai, SR calcium content and release amplitude, and contractile force. The results thus support a functional link between increased muscle performance following exercise and the operation of store-operated Orai channels at newly formed SR-TT contacts. This work has important implications for understanding muscle function and morphological adaptations to exercise as well as the newly emerging roles of store-operated calcium channels in excitable cells.

**Decision letter after peer review:**

[Editors’ note: the authors were asked to provide a plan for revisions before the editors issued a final decision. What follows is the editors’ letter requesting such plan.]

Thank you for sending your article entitled "Transverse tubule remodeling enhances Orai1-dependent Ca^2+^ entry in skeletal muscle" for peer review at *eLife*. Your article is being evaluated by three peer reviewers, and the evaluation is being overseen by a Reviewing Editor and Ronald Calabrese as the Senior Editor.

Given the list of essential revisions, including new experiments, the editors and reviewers invite you to respond with an action plan and timetable for the completion of the additional work. We plan to share your responses with the reviewers and then issue a binding recommendation.

All three reviewers thought the manuscript was an important extension of your earlier work on muscle calcium entry units (CEUs) that form after exercise. The time course of changes in SR-TT contacts, SOCE, Ca^2+^ transients, and force production following exercise provides some additional support for the idea that CEUs are involved in exercise-dependent effects on muscle contractility. However, to represent a significant advance beyond the previous work the reviewers all agreed that more quantitative experimental data are needed to establish a mechanistic link between increased SOCE and altered muscle function.

The critical question is how increased SOCE leads to the enhancement of peak Ca^2+^ transients and force. Increased SOCE cannot directly account for this because the flux is orders of magnitude smaller than the SR release flux. However, several non-exclusive mechanisms could contribute:

1) An increase in SR Ca^2+^ content leading to increased Ca^2+^ release flux might be responsible but is inconsistent with the 25% reduction in content that was reported. The method used to measure total SR content (Ca^2+^ release cocktail containing ionomycin) is not sufficiently quantitative and may also release Ca^2+^ from other compartments (e.g., mitochondria). To address this concern, a ratiometric Ca^2+^ indicator should be targeted to the SR to directly measure the store content during stimulation, and the Ca^2+^ release flux can be estimated from cytosolic transients assuming generally accepted cytosolic buffering parameters.

2) SOCE may elevate cytosolic Ca^2+^ enough to reduce fast buffering and thereby potentiate the effect of Ca^2+^ release. This could be assessed by a more detailed measurement of cytosolic Ca^2+^ between transients and estimating the effect of this increased Ca^2+^ on the buffering power.

3) Finally, SERCA may be inhibited after exercise due to glycogen depletion. Reduction of SERCA activity could enhance the transient amplitude, as well as contribute to the partial depletion of the SR and tonic SOCE that was observed. SERCA activity should be measured from the Ca^2+^ transient decay rates at different times following exercise. Importantly, such measurements may reveal a time-dependent change in Ca^2+^ handling after exercise that is not directly related to the assembly and disassembly of CEUs.

[Editors’ note: formal revisions were requested, following approval of the authors’ plan of action.]

Thank you for submitting your article "Transverse tubule remodeling enhances Orai1-dependent Ca^2+^ entry in skeletal muscle" for consideration by *eLife*. Your article has been reviewed by three peer reviewers, one of whom is a member of our Board of Reviewing Editors, and the evaluation has been overseen by Ronald Calabrese as the Senior Editor. The following individuals involved in review of your submission have agreed to reveal their identity: Eduardo Rios (Reviewer #2).

After the reviewers discussed the reviews with one another, the Reviewing Editor sent the list of essential revisions to you to verify that you could complete these within about two months. Your revision plan and timeline meet the requirements, so please proceed with the revisions and additional experiments. The complete review is printed below. Please note that additional comments have been added based on the revision plan you sent.

Summary:

Michelucci et al. present evidence that acute exercise promotes the association of sarcoplasmic reticulum (SR) stacks with extended transverse tubules (TT) in skeletal muscle, and that these contacts enhance store-operated calcium entry (SOCE) to optimize Ca^2+^ signaling and muscle contraction. This paper extends upon previous work by the authors describing the formation of SR stacks, TT extensions and increased STIM1-Orai1 colocalization (called calcium entry units, or CEUs) in the I band following exercise. The authors previously proposed that the CEUs promote resistance to fatigue during repetitive stimulation. Here the authors provide new functional evidence for this mechanism by showing that the time course of development and reversal of SR stack-TT structures parallels the timing of changes in SOCE, SR Ca^2+^ content, SR Ca^2+^ release transients, and contractile force following exercise. Using knockout mice, the authors also show that Orai1 is required for these changes, presumably through interaction with STIM1 and/or STIM2.

Essential revisions:

All three reviewers thought the manuscript is an important extension of your earlier work on muscle calcium entry units (CEUs) formed after exercise. The time course of changes in SR-TT contacts, SOCE, Ca^2+^ transients, and force production following exercise provides indirect support for the idea that CEUs are involved in exercise-dependent effects on muscle contractility. However, there was also strong agreement that more quantitative experimental data are needed to show how the observed changes in SOCE lead to altered muscle function during exercise. This could occur by several non-exclusive mechanisms, which need to be tested:

1) An increase in SR Ca^2+^ content leading to increased Ca^2+^ release flux might be responsible. The free Ca^2+^ in the SR needs to be measured during stimulus trains to determine if it is elevated. The Ca^2+^ release cocktail method will indicate total stored Ca^2+^, not free Ca^2+^. Instead, a ratiometric Ca^2+^ indicator such as D1ER should be targeted to the SR to directly measure the free luminal Ca^2+^ during stimulation.

2) SOCE may elevate cytosolic Ca^2+^ enough to reduce fast buffering and thereby potentiate the effect of Ca^2+^ release. The effect on buffering could be assessed by a more detailed measurement of cytosolic Ca^2+^ between individual transients and after stimulus trains in control and exercised mice.

3) SERCA may be inhibited after exercise due to glycogen depletion. Reduction of SERCA activity could enhance the transient amplitude, as well as contribute to the partial depletion of the SR and tonic SOCE that was observed in resting muscle after exercise. Kinetic analysis of cytosolic Ca^2+^ twitch transients (fast and slow components) can be used to estimate changes in SERCA activity and Ca^2+^ buffering from control and exercised mice before and after 10 stimulus trains. Importantly, such measurements may reveal a time-dependent change in Ca^2+^ handling after exercise that is not directly related to the assembly and disassembly of CEUs.

4) The idea that CEUs are responsible for the constitutive SOCE after exercise is not well supported. In the Discussion section, the authors suggest that the new CEUs can account for the increased tonic SOCE after exercise. The supporting argument is that "non-depletion" quench after exercise (1200 cps) + TG/CPA quench before exercise (1700 cps) = TG/CPA quench rate after exercise (2800 cps). However, the quench after exercise is driven by only partial SR depletion, so TG/CPA should add more than 1200. This argument does not make quantitative sense to me. Unless it can be better substantiated, it should be deleted.

5) A related issue concerns the idea that CEUs are solely responsible for the SOCE that augments muscle function after exercise. The loss of SR Ca^2+^ content peaks at the same time (<1 hour) as "non-depletion" SOCE, Ca^2+^ release transients and force production during trains. This complicates things. Because of tonic depletion, it is possible that pre-existing STIM and Orai complexes (activated by tonic store depletion) distinct from the new CEUs are being activated and contribute to augmented function. This possibility needs to be explicitly acknowledged, and the conclusions regarding CEU functions toned down.

[Editors' note: further revisions were requested prior to acceptance, as described below.]

Thank you for resubmitting your work entitled "Transverse tubule remodeling enhances Orai1-dependent Ca^2+^ entry in skeletal muscle" for further consideration at *eLife*. Your revised article has been favorably evaluated by Ronald Calabrese (Senior Editor) and a Reviewing Editor.

The manuscript has been improved, and the new data have addressed several of the issues that were raised in the original review. However, there are some remaining issues that need to be addressed before acceptance, as outlined below:

1) (Subsection “Total Releasable Ca^2+^ Store Content is Reduced <1 Hour After Exercise, but Increased Following Repetitive, High-frequency Stimulation” and Discussion section). After exercise, the constitutive SOCE is greatly increased, but the new D1ER measurements show that resting free Ca^2+^ in the SR is not altered. Because STIM1 and SOCE are controlled by free Ca^2+^ in ER/SR, and not the total Ca^2+^ content, how can Orai activity be increased without a reduction of free SR Ca^2+^? The decrease in total SR Ca^2+^ that was observed cannot explain the appearance of constitutive SOCE. This conundrum needs to be clearly stated and possible explanations provided. While uncertainty in the D1ER measurement might mask a small depletion of free Ca^2+^, this is unlikely to be the whole explanation since the SOCE amplitude was so large (i.e., about the same as maximal SOCE from full SR depletion before exercise).

2) The new twitch analyses are convincing, but it is not clear to anyone but an expert muscle physiologist how these were used to draw conclusions about Ca^2+^ binding and SERCA activity. In Subsection “Total Releasable Ca^2+^ Store Content is Reduced <1 Hour After Exercise, but Increased Following Repetitive, High-frequency Stimulation”, the statement that kinetics of twitch Ca^2+^ transient decay do not change needs to be explained; why was this experiment done, and what is the interpretation? In the Discussion section, it would help to discuss the possibilities that were brought up in review, including saturation of buffers and depletion of glycogen and effects on SERCA, and how the new results argue against these alternative explanations. An example of a transient should be shown along with the analysis in Figure 5—figure supplement 3 to illustrate the fast and slow components and how they were fitted.

---

## [Author Response]

[Editors’ note: what follows is the authors’ plan to address the revisions.]

All three reviewers thought the manuscript was an important extension of your earlier work on muscle calcium entry units (CEUs) that form after exercise. The time course of changes in SR-TT contacts, SOCE, Ca^2+^ transients, and force production following exercise provides some additional support for the idea that CEUs are involved in exercise-dependent effects on muscle contractility. However, to represent a significant advance beyond the previous work the reviewers all agreed that more quantitative experimental data are needed to establish a mechanistic link between increased SOCE and altered muscle function.The critical question is how increased SOCE leads to the enhancement of peak Ca^2+^ transients and force. Increased SOCE cannot directly account for this because the flux is orders of magnitude smaller than the SR release flux.

We thank the Reviewing Editor and three reviewers for their acknowledgment of the importance of the current work in supporting the conclusion that calcium entry units (CEUs) are involved in exercise-dependent effects on muscle contractility. We also appreciate the question raised regarding how increased SOCE after exercise enhances peak Ca^2+^transients and force during repetitive stimulation. Before describing our plan to address this issue, we would like to reiterate the backstory, motivation and objectives of the current study. Our initial report describing exercise-induced formation of CEUs in muscle (Boncompagni et al., 2017) concluded that CEUs formed following exercise enhance SOCE to optimize Ca^2+^release for muscle force production during repetitive stimulation. However, the implication of CEUs on SOCE and Ca^2+^release was based on indirect evidence (ICC and immuno-gold EM studies showing that exercise increases STIM1-Orai1 localization within CEUs and force production during repetitive stimulation). Direct evidence to support a role for CEUs in promoting Orai1- dependent SOCE and Ca^2+^release were lacking. Thus, a primary objective of the current study was to quantify Orai1-dependent SOCE and myoplasmic calcium transients during repetitive stimulation under conditions that result in CEU formation and disassembly. EM studies (in both EDL and FDB muscles) revealed that CEUs form < 1 hour after exercise and that TTs retract from the CEU earlier than SR stacks disassemble. We took advantage of this time course to assess changes in SOCE (Mn quench), calcium (mag-fluo4), and force (ex vivo EDL contractility) at different times during the assembly/disassembly of CEU component parts. Our results provide the first direct evidence that Orai1-dependent SOCE and Ca^2+^ release during repetitive stimulation are increased when TTs are associated with SR stacks. Unexpectedly, we found that acute exercise also results in substantial constitutive Orai1-dependent SOCE (with a magnitude similar to of maximal SOCE in control after full store depletion). These findings provide critical needed evidence to support the assertion that acute exercise promotes formation of CEUs to augment Orai1-dependent SOCE in order to optimize Ca^2+^ release and muscle contractility during repetitive stimulation.

We agree with the Editor and reviewers that the next important question to address is to explain how increased SOCE after exercise optimizes the myoplasmic Ca^2+^transient during repetitive stimulation. Since we did not have direct evidence for a specific mechanism, we refrained from speculating on this in the Discussion section of the manuscript. If provided an opportunity to submit a revised version of the manuscript, we would be happy to consider these potential mechanisms and their implications in the Discussion section. We have also initiated new experiments (outlined below) in hopes of being able to provide at least some evidence to support/exclude these possibilities. If successful, we are prepared to include the results on these studies in a revised version of the manuscript. However, given the importance and extent of the results/contributions of the study outlined above (none of which were questioned in the Editor’s summary of reviewer comments), we would hope that publication of our study will not be inexorably linked to our providing compelling evidence for an answer to this question.

We plan to revise the Discussion section of the manuscript to provide a comprehensive description of potential reasons for how increased SOCE function after acute exercise could result in enhanced peak Ca^2+^ transients during repetitive stimulation. Our discussion would include each of the three points below raised by the Editor and reviewers. Provided sufficient time, we also compete the series of experiments below designed to provide supportive evidence for one or more of these explanations.

However, several non-exclusive mechanisms could contribute:1) An increase in SR Ca^2+^ content leading to increased Ca^2+^ release flux might be responsible but is inconsistent with the 25% reduction in content that was reported. The method used to measure total SR content (Ca^2+^ release cocktail containing ionomycin) is not sufficiently quantitative and may also release Ca^2+^ from other compartments (e.g., mitochondria). To address this concern, a ratiometric Ca^2+^ indicator should be targeted to the SR to directly measure the store content during stimulation, and the Ca^2+^ release flux can be estimated from cytosolic transients assuming generally accepted cytosolic buffering parameters.

To clarify, our measurements of a ~25% reduction in total Ca^2+^ store content after acute exercise were conducted on resting non-stimulated fibers, whereas the increase in Ca^2+^release during repetitive high-frequency stimulation was only observed after 8-10 consecutive stimulation trains. In addition, the increase in peak Ca^2+^transient amplitude in fibers from exercised mice developed during the time course of each 500 ms stimulus (i.e. the peak was similar at the beginning of the train but larger by the end of the train – see blue trace in Figure 5A, middle). Thus, the evolution of the increase in evoked Ca^2+^ release is dynamic, making resting steady state store content measurements irrelevant Ca^2+^ regard to SR Ca^2+^ available for release at the end of the 10th stimulus train. Thus, one possibility is that in fibers from exercised mice, increased SOCE during each stimulus train enhances store Ca^2+^such that Ca^2+^load is increased to a level sufficient to augment Ca^2+^flux during the 10^th^ and subsequent trains. We are planning experiments where we will measure either total C Ca^2+^ a store content (using ionomycin/CPA/EGTA or ICE as before) or just RYR1-releasable store content (using 4-chloro-m-cresol) after delivery of 10 stimulus trains. Since the longitudinal SR (with SERCA) and CEU junctions (with STIM1-Orai1) lie outside the triad and lack RYR1 release channels, we expect that the ICE assay will provide a more accurate assessment of the impact of SOCE on total Ca^2+^ store content. However, it will be interesting to see if the RYR1-releasable Ca^2+^ pool following 10 tetani is increased in fibers from exercised mice. Given the extent of SR and amount of SR Ca^2+^binding proteins in skeletal muscle, the peak ICE response in experiments is dominated by Ca^2+^within the SR compartment. Consistent with this, we previously used the Ca^2+^dependent UV absorbance spectra of BAPTA to show that the vast majority (~80%) of total Ca^2+^in EDL muscle resides within the SR (Lambolay, et al., J Gen Physiol, 2015; commented on by Manno and Rios, J Gen Physiol, 2015). In addition, the peak ICE response in fura-2ff-loaded FDB fibers is markedly reduced (~90%) in fibers from CASQ1 KO mice (see Author response image 1 below). Thus, Ca^2+^within mitochondria and other compartments contributes <10% of the ICE response in these experiments. Nevertheless, we plan to repeat the store content experiments in FDB fibers expressing a low-affinity, SR targeted Ca^2+^probe (D1ER, Reggiani et al., 2010; RCepia1-ER, Suzuki et al., 2014) introduced via electroporation.

**Author response image 1. respfig1:** Effect of calsequestrin type 1 knockout (CASQ1-null) on total releasable Ca^2+^ store content. A) Representative fura-FF ratio traces of total Ca^2+^ store content elicited during application of ICE in FDB fibers from wild-type (WT) and CASQ1-null mice. B) Peak change in fura-FF ratio during ICE application. n = number of FDB fibers analyzed; *p<0.05. Data are shown as mean ± SEM.

2) SOCE may elevate cytosolic Ca^2+^ enough to reduce fast buffering and thereby potentiate the effect of Ca^2+^ release. This could be assessed by a more detailed measurement of cytosolic Ca^2+^ between transients and estimating the effect of this increased Ca^2+^ on the buffering power.

We agree. Consistent with our observation of significant constitutive SOCE after exercise, resting Ca^2+^levels were increased in FDB fibers from exercised mice (Figure 5E). Thus, the increased SOCE during repetitive stimulation could further elevate myoplasmic Ca^2+^levels enough to reduce fast Ca^2+^ buffering (e.g. parvalbumin) during subsequent stimuli. In fact, this could in part explain the time-dependent increase observed in the peak of the Ca^2+^transient during a given stimulus train in fibers from exercised mice (see blue trace in Figure 5A, middle). To further test this idea, we plan to quantify resting Ca^2+^levels in fibers from control and exercised mice immediately after 10 consecutive high-frequency stimulus trains.

3) Finally, SERCA may be inhibited after exercise due to glycogen depletion. Reduction of SERCA activity could enhance the transient amplitude, as well as contribute to the partial depletion of the SR and tonic SOCE that was observed. SERCA activity should be measured from the Ca^2+^ transient decay rates at different times following exercise. Importantly, such measurements may reveal a time-dependent change in Ca^2+^ handling after exercise that is not directly related to the assembly and disassembly of CEUs.

This is an interesting suggestion that we had not considered previously. To obtain an estimate of SERCA activity in intact fibers from control and exercised mice before and after repetitive stimulation, we will fit the decay phase of electrically-evoked twitch Ca^2+^transients measured using the low affinity Ca^2+^dye, mag-fluo-4. The decay of the Ca^2+^transient is well-described by a double exponential fit in which the fast component primarily reflects binding of Ca^2+^to fast Ca^2+^buffers (e.g. dye, parvalbumin, troponin C; Carroll et al., 1997; Baylor and Hollingsworth, 2003; Capote et al., 2005) while the slow component reflects SERCA-mediated Ca^2+^ reuptake. The relative contribution of the slow (A_slow_) and fast (A_fast_) amplitudes will be calculated by dividing the absolute value of each component by the sum of the two components (A_slow_/A_total_ and A_fast_/A_total_). This kinetic analysis will be completed on twitch Ca^2+^transients measured in FDB fibers from control and exercised mice both immediately before or after delivery of 10 consecutive high frequency stimulation trains. Comparison of the fast and slow amplitudes (absolute and relative) and the time constants of the two kinetic components will provide an index of the effects of exercise and repetitive stimulation on fast Ca^2+^buffering and SERCA activity.

[Editors’ notes: the authors’ response after being formally invited to submit a revised submission follows.]

Essential revisions:All three reviewers thought the manuscript is an important extension of your earlier work on muscle calcium entry units (CEUs) formed after exercise. The time course of changes in SR-TT contacts, SOCE, Ca^2+^ transients, and force production following exercise provides indirect support for the idea that CEUs are involved in exercise-dependent effects on muscle contractility. However, there was also strong agreement that more quantitative experimental data are needed to show how the observed changes in SOCE lead to altered muscle function during exercise. This could occur by several non-exclusive mechanisms, which need to be tested:1) An increase in SR Ca^2+^ content leading to increased Ca^2+^ release flux might be responsible. The free Ca^2+^ in the SR needs to be measured during stimulus trains to determine if it is elevated. The Ca^2+^ release cocktail method will indicate total stored Ca^2+^, not free Ca^2+^. Instead, a ratiometric Ca^2+^ indicator such as D1ER should be targeted to the SR to directly measure the free luminal Ca^2+^ during stimulation.

To clarify, our measurements of a ~25% reduction in total Ca^2+^ store content after acute exercise were conducted on resting non-stimulated fibers, whereas the increase in Ca^2+^ release during repetitive high frequency stimulation was only observed after the seventh stimulation train. In addition, the increase in peak Ca^2+^ transient amplitude in fibers from mice <1 hour after exercise developed progressively over time during the stimulation trains (i.e. peak Ca^2+^ transient amplitude did not increase significantly until the seventh stimulus train – see original Figure 5B). Thus, the evolution of the increase in evoked Ca^2+^ release is dynamic during repetitive stimulation, making resting steady-state Ca^2+^ store content measurements irrelevant with regard to SR Ca^2+^ available for release during repetitive stimulation.

Thus, one possibility is that in fibers from mice <1 hour after exercise, increased SOCE during each stimulus train progressively enhances total Ca^2+^ store content such that the total releasable Ca^2+^ load is increased to a level sufficient to augment Ca^2+^ flux during the seventh and subsequent stimulus trains. To test this idea, we conducted new experiments in fura-FF-loaded fibers to assess total releasable Ca^2+^ store content (using ionomycin/CPA/EGTA or ICE as before) after delivery of 10 stimulus trains (500 ms, 50 Hz, every 2.5 seconds). Results from these experiments are quite informative as both myoplasmic resting and total releasable Ca^2+^ store content were increased significantly after 10 stimulus trains in fibers from acutely exercised mice compared to that observed for fibers from both sedentary mice and <1 hour after exercise in the absence of stimulation (see Author response image 2). These results are consistent with the increase in Ca^2+^ transient amplitude following the seventh stimulation train, resulting at least in part from increased SOCE activity in fibers from mice <1 hour after exercise enhancing total SR Ca^2+^ store content such that releasable Ca^2+^ load is increased to a level sufficient to augment Ca^2+^ flux. These new data are included in revised Figure 5D and E (original Figure 5D and E panels are now presented as Figure 5—figure supplement 1).

**Author response image 2. respfig2:** Effect of repetitive, high-frequency stimulation on total releasable Ca^2+^ store content and resting myoplasmic Ca^2+^ concentrationin FDB fibers from control and <1 hour after exercise mice. A) Quantitative analysis of total Ca^2+^ store content elicited during application of ICE (10 µM ionomycin, 30 µM CPA, and 100 µM EGTA) in fura-FF-loaded FDB fibers in the absence (solid bars) and following (shaded bars) delivery of 10 stimulus trains (500 ms, 50 Hz, every 2.5 seconds). E) Quantitative analysis of resting myoplasmic Ca^2+^ concentration recorded in fura-2-loaded FDB fibers in the absence (solid bars) and following (shaded bars) delivery of 10 stimulus trains (500 ms, 50 Hz, every 2.5 seconds). n = number of FDB fibers analyzed; *p<0.05. Data are shown as mean ± SEM.

Given the extent of SR and level of SR Ca^2+^ binding proteins in skeletal muscle, the peak response to application of Ca^2+^ free solution containing ionomycin/cyclopiazonic acid/EGTA (ICE) is dominated by Ca^2+^ within the SR compartment. Consistent with this, we previously used the Ca^2+^ dependent UV absorbance spectra of BAPTA to show that the vast majority (~80%) of total Ca^2+^ in EDL muscle resides within the SR (Lambolay et al., 2015; commented on by Manno and Rios, 2015). Importantly, the peak ICE response in fura-FF-loaded fibers is markedly reduced (~90%) in FDB fibers from calsequestrin-1 knockout (CASQ1-null) mice, as expected since CASQ1 is the primary SR Ca^2+^ binding protein in muscle (see Author response image 1). These data suggest that Ca^2+^ within mitochondria and other compartments contributes <10% of the ICE response in these experiments.

Nevertheless, we conducted experiments using a SR-targeted Ca^2+^probe (D1ER). However, it is important to point out upfront that the D1ER probe reports the “free Ca^2+^concentration” within the SR/ER lumen, not the total releasable Ca^2+^content as assessed in our fura-FF studies. For these experiments, the D1ER probe was expressed in FDB fibers via electroporation (Rudolf et al., 2006; Canato et al., 2010). As reported previously, the D1ER probe localizes nicely to both triad junctions and longitudinal extensions of the SR, which are plentiful within the I band of the sarcomere (Figure 5—figure supplement 2A). Using identical excitation and emission gains, the D1ER YFP/CFP ratio was not significantly different in fibers from control mice and mice <1 hour after exercise both under resting conditions with a full SR Ca^2+^ store and after depletion of SR Ca^2+^ with ICE (Figure 5—figure supplement 2B and C). Thus, the spatially-averaged Ca^2+^ concentration within the SR lumen is not significantly altered by acute exercise (though areas of local depletion are possible). These findings are consistent with the strong buffering power in the SR in skeletal muscle and prior observations (e.g. Sztretye et al., 2011; Manno et al., 2013) that the free Ca^2+^ concentration in the SR lumen (~500 μM) remains largely unaltered in face of marked changes in total SR Ca^2+^ content. These results are included in the revised manuscript (Figure 5—figure supplement 2).

2) SOCE may elevate cytosolic Ca^2+^ enough to reduce fast buffering and thereby potentiate the effect of Ca^2+^ release. The effect on buffering could be assessed by a more detailed measurement of cytosolic Ca^2+^ between individual transients and after stimulus trains in control and exercised mice.

As shown above, resting myoplasmic Ca^2+^ levels were significantly elevated in FDB fibers isolated from mice <1 hour after acute exercise. Importantly, we now show (see panel B in Author response image 2) that this increase in resting Ca^2+^ in FDB fibers after acute exercise (where SOCE activity is increased) was even greater immediately after delivery of 10 stimulus trains (500 ms, 50 Hz, every 2.5 seconds). Thus, the elevation in resting Ca^2+^ after acute exercise could conceivably reduce fast buffering enough to potentiate the effect of Ca^2+^ release. However, detailed quantitative analysis of twitch Ca^2+^ transient decay kinetics before and after 10 stimulus trains does not support a significant change in either fast Ca^2+^ buffering or SERCA-mediated Ca^2+^ reuptake (see Figure 5—figure supplement 3). We updated the Discussion section of the revised manuscript to include a consideration of these.

3) Finally, SERCA may be inhibited after exercise due to glycogen depletion. Reduction of SERCA activity could enhance the transient amplitude, as well as contribute to the partial depletion of the SR and tonic SOCE that was observed in resting muscle after exercise. Kinetic analysis of cytosolic Ca^2+^ twitch transients (fast and slow components) can be used to estimate changes in SERCA activity and Ca^2+^ buffering from control and exercised mice before and after 10 stimulus trains. Importantly, such measurements may reveal a time-dependent change in Ca^2+^ handling after exercise that is not directly related to the assembly and disassembly of CEUs.

We thank the reviewer for this excellent suggestion. We completed the suggested experiments to quantify the kinetics of electrically-evoked twitch Ca^2+^ transient decay using a low affinity Ca^2+^ dye (mag-fluo-4) before and after delivery of 10 consecutive stimulus trains (500 ms, 50 Hz, every 2.5 seconds) in FDB fibers from both sedentary mice and mice <1 hour after exercise. Using this approach, the decay phase of the electrically-evoked mag-fluo-4 twitch transient is well-described by a double exponential fit in which the fast component primarily reflects Ca^2+^ binding to fast Ca^2+^ buffers while the slow component is dominated by SERCA-mediated SR Ca^2+^ reuptake (Carroll et al., 1997; Baylor and Hollingsworth, 2003; Capote et al., 2005). For these experiments, we first recorded 10 electrically-evoked twitch transients (0.5 Hz) in FDB fibers from either sedentary mice or mice <1 hour after exercise before delivery of 10 stimulus trains (500 ms, 50 Hz, every 2.5 seconds). After the 10 stimulus trains, a second set of 10 electrically-evoked twitch transients (0.5 Hz) were recorded. Despite a significant increase in the peak Ca^2+^ transient amplitude during the 10^th^ stimulus train (1.51 ± 0.07) compared to the first stimulus train (1.32 ± 0.09) only in fibers from mice <1 hour after exercise, no significant changes in Ca^2+^ transient decay kinetics were observed compared to that obtained for fibers isolated from sedentary mice. These results are included in the revised manuscript (Figure 5—figure supplement 3).

4) The idea that CEUs are responsible for the constitutive SOCE after exercise is not well supported. In the Discussion section, the authors suggest that the new CEUs can account for the increased tonic SOCE after exercise. The supporting argument is that "non-depletion" quench after exercise (1200 cps) + TG/CPA quench before exercise (1700 cps) = TG/CPA quench rate after exercise (2800 cps). However, the quench after exercise is driven by only partial SR depletion, so TG/CPA should add more than 1200. This argument does not make quantitative sense to me. Unless it can be better substantiated, it should be deleted.

As suggested, we removed this speculation from the Discussion section of the revised manuscript.

5) A related issue concerns the idea that CEUs are solely responsible for the SOCE that augments muscle function after exercise. The loss of SR Ca^2+^ content peaks at the same time (<1 hour) as "non-depletion" SOCE, Ca^2+^ release transients and force production during trains. This complicates things. Because of tonic depletion, it is possible that pre-existing STIM and Orai complexes (activated by tonic store depletion) distinct from the new CEUs are being activated and contribute to augmented function. This possibility needs to be explicitly acknowledged, and the conclusions regarding CEU functions toned down.

We thank the reviewer for pointing out this possibility, which is now explicitly acknowledged in the Discussion section of the revised manuscript.

[Editors' note: further revisions were requested prior to acceptance, as described below.]

The manuscript has been improved, and the new data have addressed several of the issues that were raised in the original review. However, there are some remaining issues that need to be addressed before acceptance, as outlined below:1) (Subsection “Total Releasable Ca^2+^ Store Content is Reduced <1 Hour After Exercise, but Increased Following Repetitive, High-frequency Stimulation” and Discussion section). After exercise, the constitutive SOCE is greatly increased, but the new D1ER measurements show that resting free Ca^2+^ in the SR is not altered. Because STIM1 and SOCE are controlled by free Ca^2+^ in ER/SR, and not the total Ca^2+^ content, how can Orai activity be increased without a reduction of free SR Ca^2+^? The decrease in total SR Ca^2+^ that was observed cannot explain the appearance of constitutive SOCE. This conundrum needs to be clearly stated and possible explanations provided. While uncertainty in the D1ER measurement might mask a small depletion of free Ca^2+^, this is unlikely to be the whole explanation since the SOCE amplitude was so large (i.e., about the same as maximal SOCE from full SR depletion before exercise).

We thank the reviewer for raising this important point.

The D1ER measurements indeed show that the “spatially-averaged” free Ca^2+^ concentration in the SR measured with this probe is not significantly altered after exercise (though areas of local depletion are possible). It is important to note, however, that our D1ER localization results indicate that the majority of the D1ER signal is in the triad junction (Figure 5—figure supplement 1A). Thus, the D1ER data primarily reflect the free Ca^2+^ concentration within the triad junction, which is not different after exercise. These results suggest that the constitutive SOCE observed after exercise does not originate from Orai1 channels in the triad junction. Rather, the constitutive Ca^2+^ entry more likely arises from Orai1 channels localized in the elongated TT within the I band region of the sarcomere that couples with SR-stack membranes forming CEUs, that are not well-reflected in the spatially-averaged D1ER signal. These observations support the idea that CEUs represent the site of constitutive Ca^2+^ entry after exercise. An alternative explanation is that exercise could produce a signal that directly activates Orai1 independent of store depletion (even within the triad).

While admittedly too superficial, we attempted to address this issue in the following sentence from the Discussion section of the prior version of the revised manuscript:

"Thus, the constitutive Ca^2+^ entry could be due to either a fraction of the SR store being partially depleted (Figure 5—figure supplement 1) or generation of a direct activator of Orai1 channels after exercise."

By “fraction of the SR being partially depleted” we were indirectly referring to CEUs formed after exercise. We also left open the possibility that exercise could produce a direct activator of Orai1 that does not require a reduction in the SR Ca^2+^ concentration. However, we appreciate the reviewers point and realize that our attempt at brevity missed several important salient points (discussed above). Therefore, we further revised this part of the Discussion section to more explicitly consider these issues.

2) The new twitch analyses are convincing, but it is not clear to anyone but an expert muscle physiologist how these were used to draw conclusions about Ca^2+^ binding and SERCA activity. In Subsection “Total Releasable Ca^2+^ Store Content is Reduced <1 Hour After Exercise, but Increased Following Repetitive, High-frequency Stimulation”, the statement that kinetics of twitch Ca^2+^ transient decay do not change needs to be explained; why was this experiment done, and what is the interpretation? In the Discussion section, it would help to discuss the possibilities that were brought up in review, including saturation of buffers and depletion of glycogen and effects on SERCA, and how the new results argue against these alternative explanations. An example of a transient should be shown along with the analysis in Figure 5—figure supplement 3 to illustrate the fast and slow components and how they were fitted.

We apologize for not providing a clear rationale in the Results section for why the twitch Ca^2+^ transient decay kinetics were analyzed (the rationale was presented in the Methods section). We revised the manuscript to include this rationale in the Results section, just before these data are presented (subsection “Total Releasable Ca^2+^ Store Content is Reduced”). As suggested, we also now include example twitch Ca^2+^ traces with superimposed two-exponential fits in Figure 5—figure supplement 3. We now also discuss potential effects of saturation of Ca^2+^ buffers, depletion of glycogen and SERCA activity in the Discussion section of the revised manuscript.